# Destabilization of *NOXA* mRNA as a common resistance mechanism to targeted therapies

Joan Montero[1,2,3,11], Cécile Gstalder[2,4,11], Daniel J. Kim[5], Dorota Sadowicz [2,4], Wayne Miles[6], Michael Manos[2], Justin R. Cidado[7], J. Paul Secrist[7,10], Adriana E. Tron[7], Keith Flaherty [8], F. Stephen Hodi[2], Charles H. Yoon[9], Anthony Letai [1,2], David E. Fisher[5,8,12]* & Rizwan Haq [2,4,12]*

Most targeted cancer therapies fail to achieve complete tumor regressions or attain durable remissions. To understand why these treatments fail to induce robust cytotoxic responses despite appropriately targeting oncogenic drivers, here we systematically interrogated the dependence of cancer cells on the BCL-2 family of apoptotic proteins after drug treatment. We observe that multiple targeted therapies, including BRAF or EGFR inhibitors, rapidly deplete the pro-apoptotic factor NOXA, thus creating a dependence on the anti-apoptotic protein MCL-1. This adaptation requires a pathway leading to destabilization of the *NOXA* mRNA transcript. We find that interruption of this mechanism of anti-apoptotic adaptive resistance dramatically increases cytotoxic responses in cell lines and a murine melanoma model. These results identify *NOXA* mRNA destabilization/MCL-1 adaptation as a non-genomic mechanism that limits apoptotic responses, suggesting that sequencing of MCL-1 inhibitors with targeted therapies could overcome such widespread and clinically important resistance.

[1] Division of Hematologic Neoplasia/Malignancies, Dana-Farber Cancer Institute, Harvard Medical School, 450 Brookline Ave, Boston 02115 MA, USA. [2] Department of Medical Oncology, Dana-Farber Cancer Institute, Harvard Medical School, 450 Brookline Ave, Boston 02115 MA, USA. [3] Institute for Bioengineering of Catalonia, C/Baldiri Reixac 15-21, Ed. Hèlix 3ª planta · 08028, Barcelona, Spain. [4] Division of Molecular and Cellular Oncology, Dana-Farber Cancer Institute, Harvard Medical School, 450 Brookline Ave, Boston 02115 MA, USA. [5] Department of Dermatology and Cutaneous Biology Research Center, Massachusetts General Hospital, Harvard Medical School, 44 Fruit Street, Boston, MA 02114, USA. [6] Department of Molecular Genetics, The Ohio State University, 820 Biomedical Research Tower 460 West 12th Avenue, Columbus 43210 OH, USA. [7] Bioscience, Oncology IMED Biotech Unit, AstraZeneca, 35 Gatehouse Dr, Waltham, Boston 02451 MA, USA. [8] Massachusetts General Hospital Cancer Center, Massachusetts General Hospital, Harvard Medical School, 44 Fruit Street, Boston, MA 02114, USA. [9] Department of Surgery, Brigham and Women's Hospital, Harvard Medical School, 75 Francis Street, Boston 02115, USA. [10]Present address: LifeMine Therapeutics, 100 Acorn Park Drive, 6th Floor Cambridge, Cambridge, MA 02140, USA. [11]These authors contributed equally: Joan Montero, Cécile Gstalder. [12]These authors jointly supervised: David E. Fisher, Rizwan Haq. *email: dfisher3@partners.org; rizwan_haq@dfci.harvard.edu

Constitutive activation of the mitogen-activated protein kinase (MAPK) signal transduction pathway is the most commonly dysregulated pathway in cancer[1]. In melanoma, for example, mutations in the *BRAF* protein kinase, which are found in ∼50% of tumors, drive the hyper-activation of MAPK signaling[2]. Mutations in the epithelial growth factor receptor (*EGFR*)[3,4] or other receptor tyrosine kinase family members[5,6], signal transducers (such as *RAS*)[7] similarly lead to the dysregulation of MAPK pathway in many other cancer cell types. Consistent with the requirement of MAPK for growth and survival of cancer cells, targeted therapies that suppress MAPK signaling lead to clinical responses in patients[8–10]. However, most patients experience only partial or transient responses to MAPK pathway inhibitors, suggesting that tumor cells adapt to drug treatment either through additional genomic or non-genomic changes. Biopsies from patients being treated with BRAF inhibitors several weeks after treatment with the BRAF inhibitor vemurafenib have revealed that despite potent initial inhibition of MAPK signaling and cell growth inhibition, cytotoxic responses are modest and highly variable among patients[11]. These data suggest that improving cytotoxic responses to targeted therapies could overcome widespread and clinically important resistance in multiple cancer types.

Cytotoxic responses to targeted therapy are regulated by pro- and anti-apoptotic members of the BCL-2 family[12], suggesting that their modulation could enhance targeted therapy response. Consistent with this hypothesis, reducing the expression of the BCL-2 anti-apoptotic protein MCL-1 sensitized *EGFR*-mutant non-small cell lung cancers to MAPK pathway inhibitors[13]. In *KRAS*-mutant cancer models, inhibition of the BCL-XL protein caused marked regressions of tumors when combined with MAPK pathway inhibitors[14]. We have previously shown that expression of the anti-apoptotic family member *BCL2A1* (BFL-1) inversely correlates with sensitivity to BRAF inhibitors[15].

Based on these and other data, drugs that directly target BCL-2 family proteins have been the focus of intensive pharmaceutical interest. For example, the selective anti-cancer activity of venetoclax, an inhibitor of the anti-apoptotic protein BCL-2, has finally validated the clinical utility of directly targeting tumor cell death[16–18]. Several other drugs targeting cell death pathways are in pre-clinical testing or early phase clinical trials, including recently described small molecule inhibitors of the MCL-1 anti-apoptotic protein[19]. However, such agents have thus far shown little efficacy in many cancer types, including most solid tumors[19–21]. Therefore, a key challenge to optimize the opportunity provided by these apoptosis-inducing drugs is the markedly varied responses observed among different patients[16,22].

To date, there are few robust biomarkers that identify the predisposition of a cancer cell to undergo apoptosis. Although genomic[23], transcript,[24–26] and protein levels of some cell death proteins are associated with therapeutic response, no single biomarker has so far been sufficient to predict a cell's apoptotic response to a given treatment, probably since the physical association between these proteins also is crucial[27]. Guided by the need to identify patients who may benefit from inhibitors of anti-apoptotic proteins, we have performed a sensitization genetic screen to identify the anti-apoptotic family members that limit cytotoxic responses to targeted therapies in cancer cells and primary patient samples. Here, we report that multiple inhibitors of the MAPK pathway lead to rapid changes in dependence on BCL-2 family members, indicating that adaptive changes, rather than genomic changes, underlie apoptotic resistance to targeted therapies. Mechanistically, we found that these drugs lead to the depletion of the BCL-2 family pro-apoptotic factor *NOXA* (also known as *PMAIP1*). Reduction of *NOXA* requires the destabilization of its mRNA by the RNA

decay protein ZFP3636/TTP. We find that loss of *NOXA* increases MCL-1 dependence and binding to other BAX/BAK pro-apoptotic factors such as BIM, thereby potently antagonizing the ability of the targeted agents to induce efficient apoptotic death. Conversely, interruption of this mechanism of anti-apoptotic adaptive resistance (via the use of MCL-1 inhibitors) dramatically increased cytotoxic responses in vitro and in murine melanoma models. These results identify a feedback/survival mechanism involving RNA destabilization for preventing efficient apoptotic responses to MAPK pathway inhibition following multiple targeted cancer treatments, suggesting therapeutic strategies to overcome such widespread and clinically important resistance.

## Results

**Targeted therapies induce rapid dependence on MCL-1**. To determine whether the suppression of anti-apoptotic family member(s) could enhance the activity of targeted therapies, we suppressed individual BCL-2 anti-apoptotic family members[28] using siRNA in 21 cancer cell lines of different lineages, each with a distinct, dominant driver oncoprotein (Fig. 1a; Supplementary Table 1). We treated each cell line with a small molecule inhibitor of each driver oncoprotein over 250-fold dose concentrations (40 nM to 10 μM) and measured cell number after 48 h. Specifically, we used the BRAF inhibitor PLX4720 for *BRAF*-mutant cells; imatinib for *KIT*-mutant cell lines; gefitinib for *EGFR*-mutant cells; lapatinib for *ERBB2*-mutant lines; and crizotinib for *MET*-amplified or *ALK*-mutant cells (Supplementary Table 2). Although knockdown efficiency was comparable among the different siRNAs (Supplementary Fig. 1a, b), suppression of *MCL1* strongly sensitized most cell lines, independent of lineage, driver oncoprotein, or targeted therapy (Fig. 1b). Suppression of other anti-apoptotic BCL-2 family members did not consistently affect the targeted therapy responses. To independently test the results from this screen, we treated the *BRAF*-mutant melanoma cell line A375M with siRNA targeting the non-coding region of *MCL1* (Supplementary Fig. 1c). Suppression of *MCL1* alone did not induce significant apoptosis, but concomitant treatment with the MEK inhibitor trametinib dramatically increased PARP cleavage. These effects could be rescued upon the expression of a non-targetable *MCL1* cDNA. Ectopic expression of MCL-1 also inhibited the cytotoxicity of BRAF inhibitors at higher doses (Supplementary Fig. 1d), collectively demonstrating that MCL-1 is both necessary and sufficient for resistance to multiple targeted therapies.

We hypothesized that the effects of MCL-1 suppression on targeted therapies might be related to either intrinsic dependence of the cell lines on MCL-1, or dependence that is generated upon drug treatment. To distinguish between these possibilities, we performed dynamic BH3 profiling (DBP) to analyze the dependence of A375M melanoma cells on each BCL-2 family member before and after treatment with BRAF or MEK inhibitors[29]. This method can determine the cells' sensitivity to anti-cancer agents and their dependence on individual anti-apoptotic BCL-2 family members within hours of drug treatment, as compared with days required for genetic suppression by siRNA[30]. As expected, the BIM peptide (which binds to all anti-apoptotic BCL-2 family proteins and can directly activate BAX and BAK) induced a significant increase in overall mitochondrial outer membrane permeabilization (MOMP) after 36 h treatment with BRAF or MEK inhibitors (Fig. 1c). Surprisingly, treatment with two peptides that specifically bind to MCL-1 (NOXA and MS1 BH3 peptides) induced an increase in MOMP when exposed to BRAF and MEK inhibitors at longer time points, indicating an increase in MCL-1 dependence. In contrast, dependence on

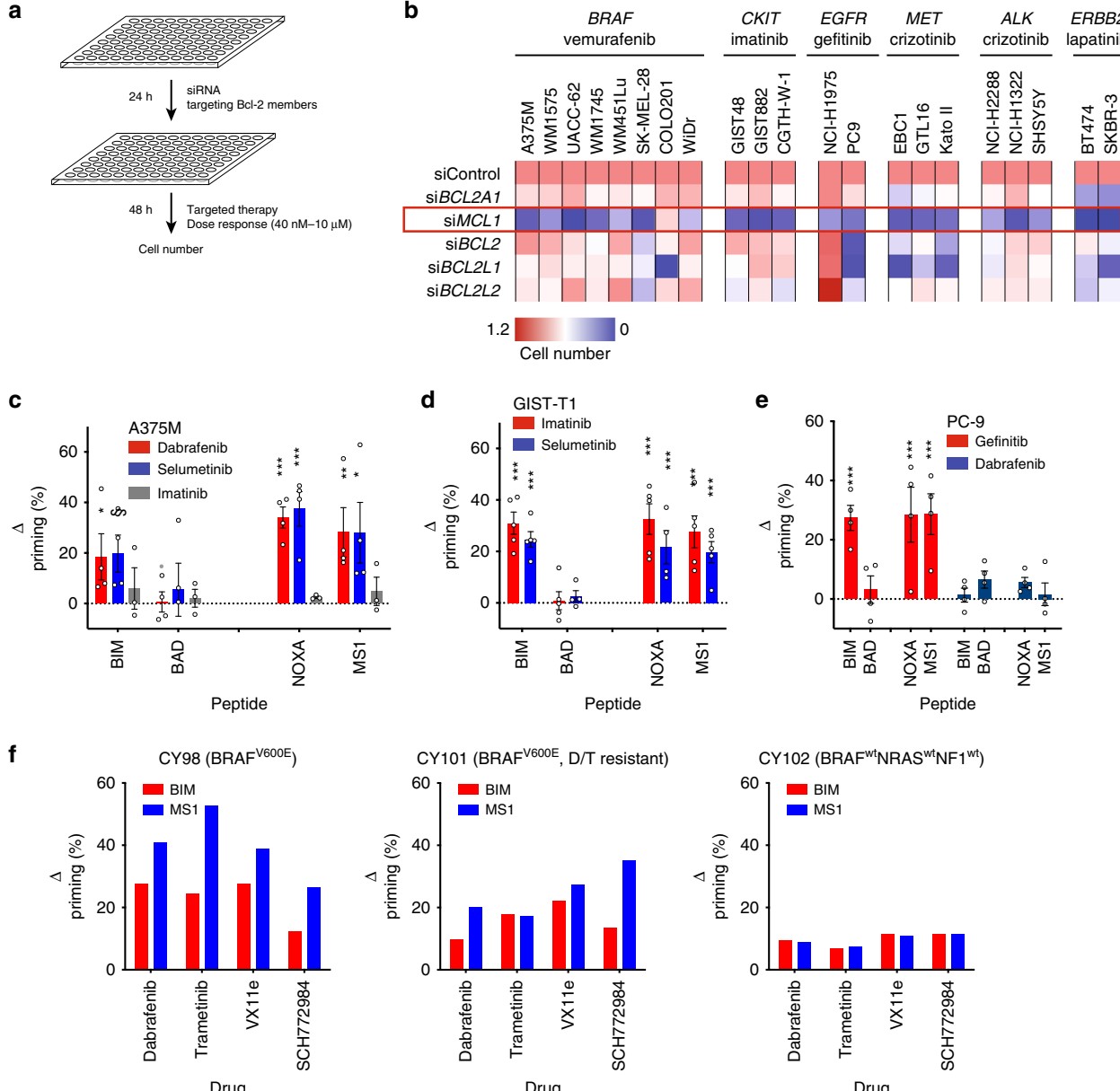

**Fig. 1** Targeted therapies induce dependence on MCL-1. **a** Scheme for sensitization siRNA screen to targeted therapies. **b** Cell number following targeting of the anti-apoptotic BCL-2 family by siRNA and targeted therapies (10 μM), relative to vehicle-treated cells. PLX4720 was used for *BRAF*-mutant cells, imatinib was used for *CKIT*-mutant cells, gefitinib was used use for *EGFR*-mutant cells; crizotinib was used for *MET*- and *ALK*-mutant cells and lapatinib was used for *ERBB2*-amplified cells. Cell number was normalized to cells transfected with control siRNA and treated with drug vehicle. **c**, **d**, **e** Dynamic BH3 profiling (DBP) of cancer cell lines following 36 h treatment with the indicated drug (1 μM). Mitochondrial permeabilization is calculated relative to vehicle-treated cells. Statistical significance (*n* = 4) determined using the Holm–Sidak method. ***, adjusted *P* value < 0.001 comparing drug treatment vs vehicle control; **, adjusted *P* value < 0.01; *, adjusted *P* value < 0.05; §, adjusted *P* value < 0.1. Comparisons with adjusted *P* value ≥ 0.1 are designated without any symbol. **f** DBP profiles following BRAF (1 μM dabrafenib), MEK (0.1 μM trametinib), and ERK inhibitors (1 μM VX11e and SCH772984) on freshly obtained cells from melanoma patients (*n* = 1). See also Supplementary Fig. 1 and Supplementary Tables 1 and 2. Source data for siRNA screen are provided as a Source Data file. Error bars indicate mean ± s.e.m. of indicated replicates

BCL-2/BCL-XL/BCL-W (measured with the BAD peptide) did not change with drug treatment. Further, no effect was seen upon treatment of A375M cells with imatinib, indicating that the results were specific to inhibition of the driver oncoprotein. Overall apoptotic response to dabrafenib could be seen within the first 24 h of exposure to dabrafenib and sustained over time (Supplementary Fig. 1e), whereas the dependence on MCL-1 was comparatively delayed, starting at 36 h, and continued to increase with time. These effects were also seen with the combination of BRAF and MEK inhibitors, the current standard of care for

melanoma targeted therapy (Supplementary Fig. 1f). However, we observed no effect of MEK inhibitors on the MCL-1 dependence of the *BRAF* wild-type cell line IPC-298 (Supplementary Fig. 1g).

To evaluate if these results are generalizable to other cancer types and primary tumor samples, we tested the effect of imatinib on the *KIT*-mutant gastrointestinal stromal tumor (GIST) cell line GIST-T1 (Fig. 1d) and the effects of gefitinib on the *EGFR*-mutant lung cancer cell line PC9 (Fig. 1e). In both cases, drug treatment induced the expected increase in overall priming (assessed with the BIM peptide) and specifically induced MCL-1

dependence (assessed with the NOXA and MS1 peptides). Interestingly, treatment of the GIST cell line with a MEK inhibitor also induced MCL-1 dependence. These data suggest that although inhibition of KIT and EGFR signaling impacts multiple downstream pathways, the induced dependence on MCL-1 may be related specifically to MAPK pathway inhibition.

We next used a similar DBP approach to evaluate if these effects were seen in freshly harvested tumor samples[29]. To distinguish melanoma cells from tumor-associated stromal and lymphoid cells that often contaminate melanoma biopsies, we introduced two modifications to this assay. First, we stained cells with antibodies that detect the expression of the melanocyte lineage-restricted cell surface protein CSPG4[31], the lymphocyte marker CD45, and the stromal marker Fibroblast-activation protein (FAP). Second, we detected mitochondrial permeabilization using a flow cytometer (FACS)-based assay[18,32] rather than the fluorometric assay. These modifications permitted selection of melanoma cells specifically (e.g., CSPG4-positive, CD45-negative, FAP-negative population) (Supplementary Fig. 1h). Treatment of BRAF-mutant primary melanoma cells (Supplementary Table 3) with up to 24 h of BRAF, MEK, or ERK inhibitors induced MCL-1 dependence (Fig. 1f). Conversely, we observed minimal change in MCL-1 dependence in cells derived from a patient with a BRAF-mutant melanoma who had developed resistance to BRAF/ MEK inhibitors. Similarly, a representative BRAF/NRAS wild-type melanoma was also not responsive to BRAF or MEK inhibitors. Collectively, we conclude that targeted therapies create a new dependence on MCL-1 in multiple cancer types and in primary tumors.

**MAPK inhibition suppresses the pro-apoptotic factor NOXA.** To investigate the mechanism of adaptive MCL-1 dependence, we next evaluated the effect of targeted therapies on anti-apoptotic and pro-apoptotic BCL-2 family by western blot (Fig. 2a). As previously reported[33], the treatment of SK-MEL-5 cells with a BRAF inhibitor led to an upregulation of BIM. Although we observed no effect of the BRAF inhibitor treatment on MCL-1 levels, there was a near-complete loss of the expression of NOXA, a protein that directly binds and inactivates MCL-1[34]. We observed similar effects upon treatment with a MEK inhibitor (Fig. 2b) and in three other BRAF-mutant melanoma cell lines with BRAF or MEK inhibitors (Supplementary Fig. 2a). The effect was independent of tissue type as it was also observed in a BRAF-mutant colorectal cancer cell line (Supplementary Fig. 2b). We also saw similar effects in an imatinib-treated KIT-mutant GIST cell line (Fig. 2c; Supplementary Fig. 2c), MET-dependent cell lines or ALK-mutant cells treated with the MET/ALK inhibitor crizotinib (Supplementary Fig. 2d, e), a HER2-amplified breast cancer cell line treated with HER2 inhibitor lapatinib (Supplementary Fig. 2f), and an EGFR-mutant lung cancer cell line treated with the EGFR inhibitor gefitinib (Supplementary Fig. 2g). This NOXA downregulation was observed with structurally distinct BRAF/MEK inhibitors but not with an inhibitor of PI3K (Supplementary Fig. 2h).

To evaluate if the effects of MAPK suppression on NOXA were related to changes in mRNA, we interrogated published microarray data of several BRAF-mutant cell lines treated with the MEK inhibitor PD325901[35] (Fig. 2d). We observed that NOXA levels were strongly downregulated in multiple cell lines upon drug treatment, whereas expression of MCL1 and all other BCL-2 family members were not significantly changed. We independently confirmed these results in A375M cells (Fig. 2e). The effects of MAPK suppression on NOXA mRNA were seen in both BRAF and NRAS mutant melanoma cell lines (Supplementary Fig. 2i). Comparison of data sets from CKIT-mutant cells

treated with imatinib, EGFR-mutant cell lines treated with gefitinib or ALK-mutant tumors treated with crizotinib also exhibited decreased levels of NOXA mRNA (Supplementary Fig. 2j). MEK inhibitor, but not BRAF inhibitor treatment of primary melanocytes also suppressed NOXA (Supplementary Fig. 2k). To evaluate if MAPK is essential for the expression of NOXA using an alternative approach, we suppressed MAPK using siRNAs targeting ERK1/2. We observed that suppression of the MAPK pathway was associated with decreased NOXA protein (Supplementary Fig. 2l) and mRNA (Supplementary Fig. 2m).

To test the functional importance of NOXA suppression in response to BRAF inhibitors, we constitutively expressed NOXA in A375M cells. Expression of NOXA did not affect the levels of MCL-1 or BIM (Fig. 2f). We next measured cell number after 24 h treatment with a BRAF inhibitor. NOXA expression by itself did not significantly affect cell number but strongly sensitized A375M cells to the drug (Fig. 2g). Similar results were observed in a mouse xenograft model of melanoma (Fig. 2h). To evaluate the relationship of NOXA and MCL-1 in primary melanocytes, we used siRNAs to suppress BCL-2 family with or without MCL1 knockdown (Supplementary Fig. 2n). Suppression of MCL1 led to decreased cell proliferation in these cells, which predictably could be rescued by depletion of BAK1. Among BH3 family proteins, only NOXA suppression could rescue the cytotoxic effects of MCL1 suppression, indicating that NOXA is essential for apoptosis triggered upon MCL1 suppression. Together, these data suggest that the MAPK pathway is necessary for the expression of a functionally important antagonist of MCL-1, NOXA.

**MAPK inhibition destabilize NOXA mRNA via TTP/ZFP36.** To mechanistically understand how MAPK suppression leads to decreases in NOXA mRNA, we monitored the expression of NOXA mRNA after 1, 2, 8, and 24 h of BRAF or MEK inhibitor treatment of A375M cells (Fig. 3a). Loss of NOXA mRNA was rapid, with a half-life of ~ 2 h, consistent with other published data showing a highly unstable mRNA[36]. This rapid loss of NOXA mRNA prompted us to evaluate whether MAPK suppression destabilized its mRNA. We treated cells with the transcriptional inhibitor actinomycin D, followed by the BRAF inhibitor PLX4720. Actinomycin D suppressed the upregulation of TRPM1 mRNA, which is transcriptionally activated upon treatment of melanoma cells with BRAF inhibitor (Supplementary Fig.3a)[37] but did not affect HIF1A mRNA (Supplementary Fig. 3b), confirming that the dose of actinomycin used was sufficient to block transcription completely. Concomitant treatment of actinomycin-treated A375M cells with PLX4720 enhanced the loss of NOXA mRNA (Fig. 3b), suggesting that suppression of NOXA by BRAF/MEK inhibitors could not be entirely owing to effects on its transcription.

To identify putative post-transcriptional regulators of NOXA, we inspected the sequence of NOXA mRNA and identified a consensus binding motif in the 3′-untranslated region for the TTP/ZFP36 RNA decay protein family (Fig. 3c)[38]. This motif has previously been shown to be sufficient for the recruitment of three distinctly encoded proteins within the ZFP36 family (ZFP36/TTP, ZFP36L1, and ZFP36L2)[39]. The TTP/ZFP36 family has previously been demonstrated to undergo MAPK-mediated phosphorylation, which leads to its destabilization and diminished enzymatic activity[40–43]. Therefore, we asked whether ZFP36 is necessary for NOXA mRNA decay upon BRAF inhibition in melanoma cells. The suppression of ZFP36 by pooled (Fig. 3d) or two independent (Supplementary Fig. 3c) siRNAs induced the expression of NOXA mRNA, whereas ZFP36L1 or ZFP36L2 had

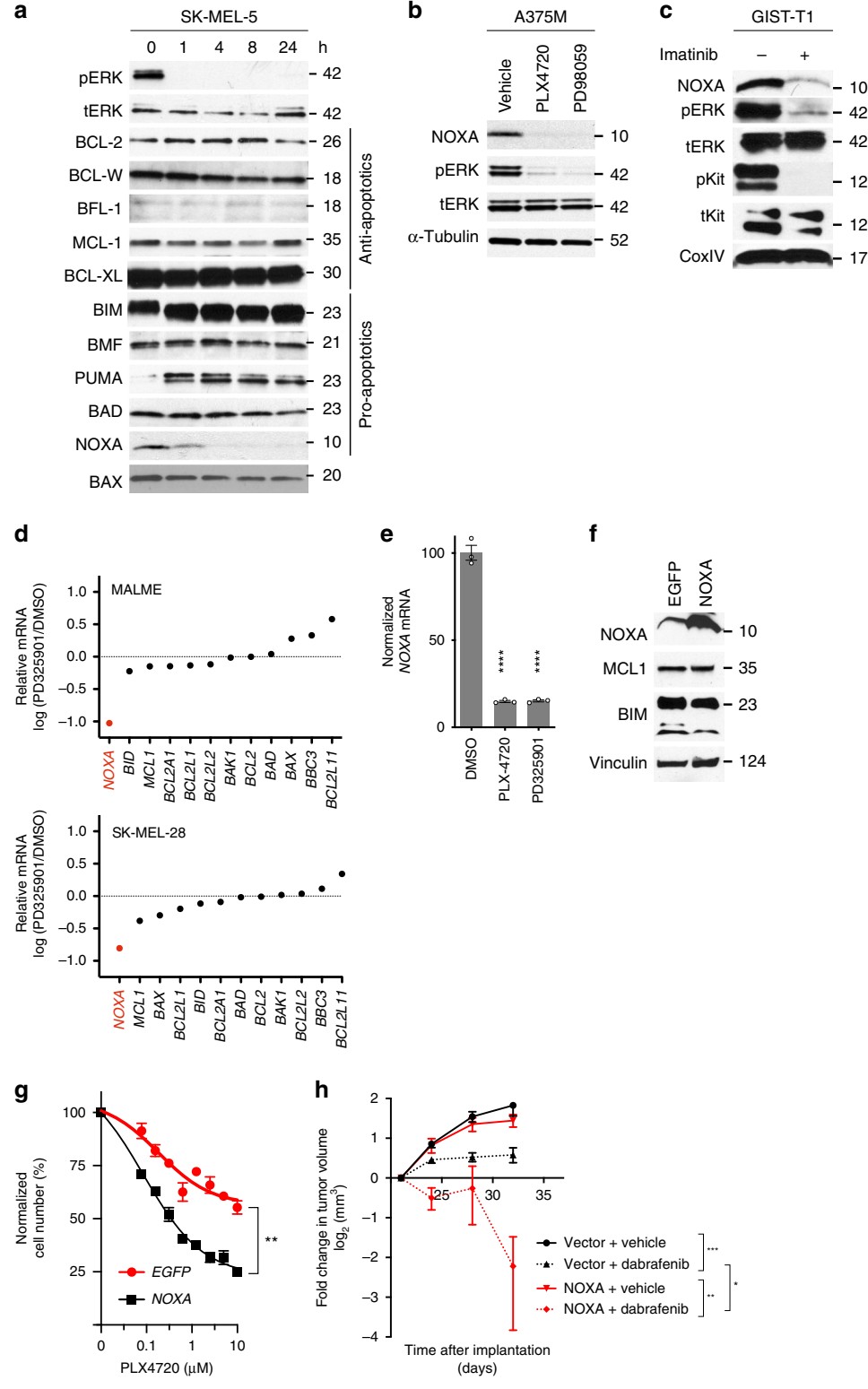

more modest effects. The induction of the NOXA mRNA was similar in magnitude to an established direct target of ZFP36, *LIF*. To evaluate whether ZFP36 was required for the decay of *NOXA* mRNA, we transfected two independent siRNAs targeting *ZFP36* into A375M cells and treated the cells with PLX4720. We noticed that ZFP36 knockdown, although incomplete (Fig. 3e), blocked the decay of *NOXA* mRNA (Fig. 3f). The effects of ZFP36

knockdown were not related to any unexpected effects of siZFP36 on the MAPK pathway (Supplementary Fig. 3d).

To evaluate if ZFP36 was sufficient to suppress *NOXA*, we overexpressed ZFP36 in several cell lines and assessed its effect on *NOXA* mRNA. Ectopic ZFP36 expression significantly suppressed *NOXA* and *JUN* (a known target of ZFP36[44]) mRNA levels but did not affect *E2F4* (an mRNA that has a short 3′-UTR and no

**Fig. 2** Targeted therapies suppress *NOXA* mRNA. **a** Protein levels of BCL-2-family members following treatment of SK-MEL-5 melanoma cells treated PLX4720 (1 μM). **b** Effect of BRAF and MEK inhibitors on NOXA protein in A375M melanoma cells. **c** Effect of imatinib on NOXA in *KIT*-mutant GIST-T1 cells. **d** Changes in BCL-2 family mRNA expression following MEK inhibitor treatment as assessed in published microarray analysis (GSE20051). For each gene, the expression is normalized to that of DMSO-treated cells. **e** Effect of BRAF and MEK inhibitor on *NOXA* mRNA in A375M cells. Statistical significance of drug versus vehicle-treated cells ($n = 3$) was determined using Holm–Sidak multiple comparison test. ****, adjusted value < 0.0001. **f** Western blot of NOXA, BIM, and MCL1 in A375M cells expressing NOXA. **g** Effect of overexpression of NOXA on number of A375M cells following PLX4720 treatment ($n = 3$ per group). Statistical significance of growth inhibition was done using extra sum-of-squares *F*-test. **, adjusted *P* value < 0.01. **h** Effect of overexpression of NOXA on the growth of A375M xenografts on response to dabrafenib ($n = 10$ per group). Tumor size shown is relative to pre-treatment (day 22). Fold change in tumor volume at day 32 was compared using ANOVA with Sidak multiple comparison tests. *, adjusted *P* value < 0.05; **, adjusted *P* value < 0.01; ***, adjusted *P* value, < 0.001. Error bars indicate mean ± s.e.m. of indicated replicates. See also Supplementary Fig. 2. Source data for all Western blots are provided as a Source Data file

known TTP-binding sites) (Fig. 3g). Similar results were seen in a GIST cell line (Supplementary Fig. 3f), another melanoma cell line (Supplementary Fig. 3g) and a gastric cancer cell line (Supplementary Fig. 3h). To evaluate whether ZFP36 protein could directly interact with the *NOXA* mRNA, we conducted immunoprecipitation of ZFP36 under RNase-free conditions (Supplementary Fig. 3i). We quantified the associated *NOXA* mRNA by qPCR (Fig. 3h). From these experiments, we found that the mRNAs for *NOXA* and the established ZFP36 target, *CCND1* were strongly enriched compared with the immunoglobulin control, demonstrating that the ZFP36 protein can directly associate with the NOXA mRNA.

TTP/ZFP36 has previously been observed to be directly phosphorylated by MAPK at serine 218 and serine 228 (Fig. 3i)[42,45,46], leading to its degradation[41] or altered subcellular localization[40]. To evaluate if clinically relevant targeted therapies impacted the stability of ZFP36, we constitutively expressed wild-type, and mutant forms of TTP lacking either phosphorylation sites (Fig. 3j). BRAF inhibitor treatment led to increased levels of wild-type ZFP36. In contrast, ZFP36 mutants lacking either target serine both had a higher basal expression of ZFP36, which did not increase upon drug treatment.

**Targeted therapies create vulnerability to MCL-1 inhibitors**. Our data indicate that MAPK suppression creates a new dependence on MCL-1, which is associated with decreased *NOXA* mRNA. As NOXA specifically binds MCL-1, we hypothesized that loss of NOXA could free NOXA-bound MCL-1 to associate with other pro-apoptotic BCL-2 family members. To evaluate this possibility, we immunoprecipitated MCL-1 from drug-treated A375M melanoma cells and measured its association with NOXA by western blotting. As expected, BRAF inhibitors led to a decreased association with NOXA (Fig. 4b) without any change in the expression of MCL-1 (Fig. 4a). As MCL-1 can also bind to both NOXA and the pro-apoptotic BAX/BAK activating factor BIM, we next evaluated the effect of drug treatment on the association of MCL-1 to BIM. BIM is essential for BAX/BAK activation and apoptotic engagement[18]. Interaction of MCL-1 with BIM was strongly increased upon treatment of A375M cells with PLX4720. We confirmed these results by immunoprecipitating BIM followed by western blotting using an MCL-1 antibody (Fig. 4c). To evaluate if these effects were seen in other cancer types, we also treated *KIT*-mutant GIST cells with imatinib. Similar to the experiments with the melanoma cells, imatinib suppressed the association of MCL-1 with NOXA and concomitantly increased its association with BIM (Supplementary Fig. 4a–c).

The increased association of MCL-1 with BIM upon treatment with targeted agents could have major implications for their cytotoxic effects as binding of MCL-1 inhibits the pro-apoptotic activity of BIM. To evaluate the functional impact of the redistribution of MCL-1 to BIM, we evaluated the effects of A1210477, an MCL-1 inhibitor, on A375M cells. A1210477 did

not affect BIM or NOXA levels but increased MCL-1 levels consistent with prior reports (Fig. 4d)[47]. A1210477 also did not affect the ability of the BRAF inhibitor dabrafenib to suppress NOXA or the MAPK pathway. We next immunoprecipitated MCL-1 after 24 h treatment with dabrafenib, A1210477, or both. Whereas dabrafenib induced the expected association of MCL-1 with BIM, concomitant treatment with A1210477 blocked this association (Fig. 4e). Identical results were seen upon immuno-precipitation with an anti-BIM antibody (Supplementary Fig. 4d).

The observation that targeted inhibitors of the MAPK pathway induce MCL-1 dependence suggests that prior treatment of cancer cells with MAPK pathway inhibitors might promote sensitivity to MCL-1 inhibitors. Several MCL-1 inhibitors are being evaluated in pre-clinical and early-stage clinical trials but how to deploy these drugs for maximal therapeutic utility, as single agents or in combination, is so far poorly defined. In fact, most cancer cell lines are resistant to the effects of MCL-1 inhibitors[19], highlighting the need to develop alternative combination therapies. To test the hypothesis that BRAF/MEK inhibitors could sensitize melanoma cells to MCL-1 inhibitors, we compared the effects of A1210477 given before versus after BRAF inhibitor treatment in A375M melanoma cells (Fig. 4f). Each drug was applied for the same amount of time, but the order of the drugs was reversed. Treatment with A1210477 followed by dabrafenib was antagonistic (Fig. 4g). In contrast, dabrafenib strongly synergized cells to the effects of A1210477. The differences in the effects of these schedules were not related to altered MAPK signaling or differential effects on NOXA protein (Supplementary Fig. 4e). To quantify the effects of these drugs on apoptosis, we treated A375M cells with dabrafenib (0–60 h) followed by AZD5991 (a specific, more potent inhibitor of MCL-1[48]) or control (16 h) and measured apoptosis by Annexin-V staining. Treatment with AZD5991 alone had virtually no effect at 16 h (Time 0; Fig. 4h) and treatment of cells with dabrafenib alone induced ≤ 30% apoptosis, even after 60 h of drug treatment. However, treatment with dabrafenib strongly sensitized cells to the MCL-1 inhibitor (88% apoptosis at 60 h). There was no effect of dabrafenib followed by AZD5991 in a *BRAF* wild-type cell (Fig. 4i). To evaluate the robustness of these observed effects, we performed similar experiments in melanoma, GIST, and lung cancer cell lines using AZD5991 or a BCL-2/BCL-XL inhibitor, AZD4320. Strong synergy was observed upon treatment of cells after suppression of the oncoprotein followed by the MCL-1 inhibitor, whereas limited synergy was seen with the BCL-2 inhibitor (Fig. 4j). Concomitant treatment with the MCL-1 inhibitor and BRAF inhibitor had modest synergy in A375M melanoma cells. Collectively, these data indicate that pre-treatment with BRAF or MEK inhibitors may sensitize to MCL-1 inhibitors, whereas the opposite schedule is significantly less effective.

**In vivo evaluation of sequential BRAF and MCL-1 inhibition**. To test the relevance of anti-apoptotic adaptation in vivo, we

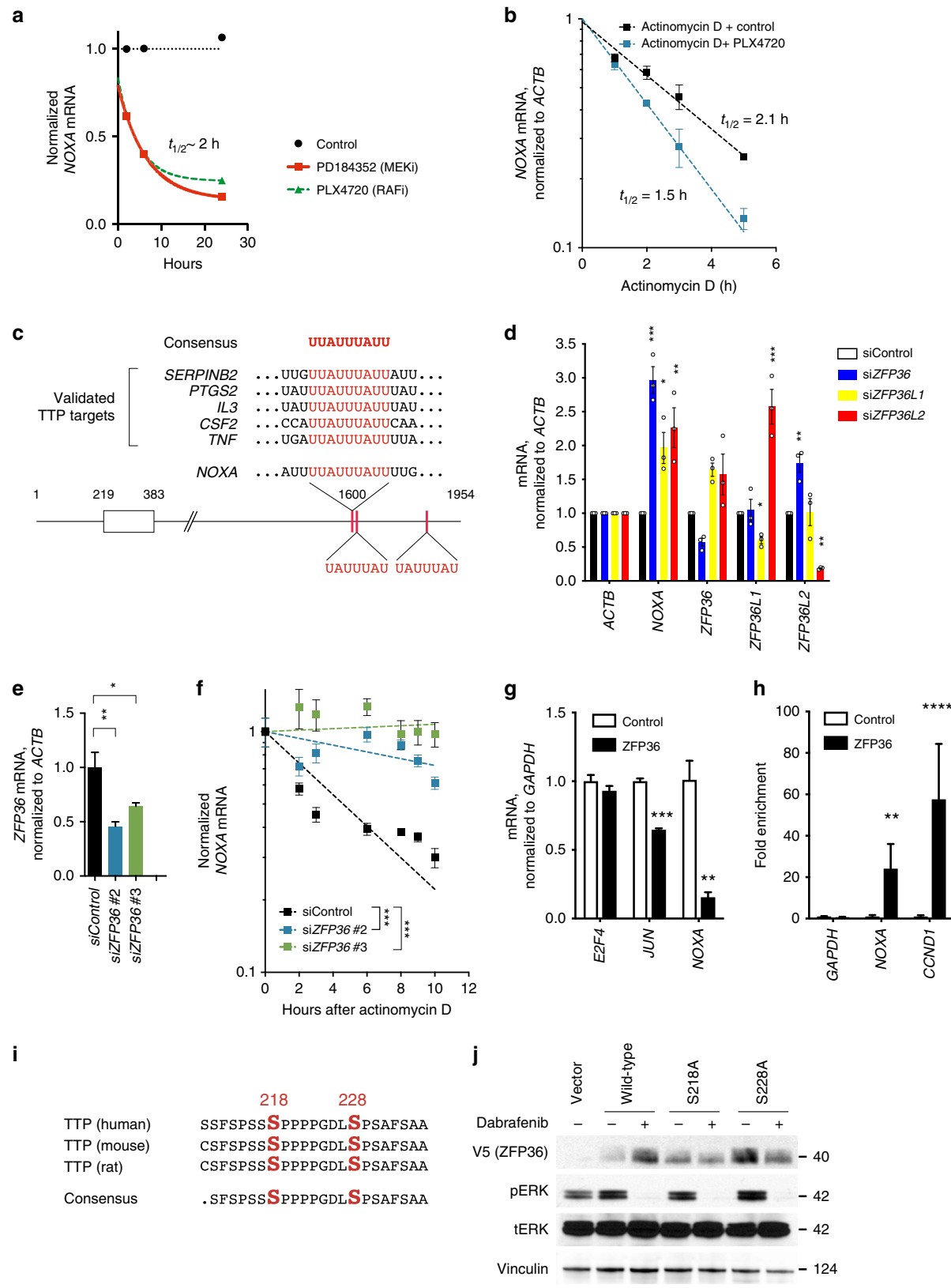

obtained RNA from three melanoma patients with *BRAF*-mutant melanomas before, and 14 days after starting single agent vemurafenib. In each case, on-treatment *NOXA* and *DUSP4* (a measure of MAPK activity) mRNA were suppressed compared

with paired pre-treatment samples (Fig. 5a), consistent with our cell line data (see Fig. 2).

Next, we tested if BRAF inhibition induced MCL-1 dependence in vivo. We injected nude mice with A375M cells and after

**Fig. 3** ERK suppression decreases NOXA expression via TTP/ZFP36. **a** Quantification of *NOXA* mRNA following treatment of BRAF-mutant A375M cells with MEK or BRAF inhibitors. **b** *NOXA* mRNA upon treatment with actinomycin D with or without BRAF inhibitor treatment ($n = 2$–3 per group). **c** Consensus sequence of binding sites for TTP/ZFP36 family proteins depicting the location of putative AU-rich sequences in *NOXA* mRNA. **d** Quantification of indicated mRNAs following transfection of A375M cells with siRNAs targeting ZFP36 family ($n = 3$). Statistical comparison was done using one-way ANOVA with Dunnett's multiple comparison test. *, adjusted *P* value < 0.05; **, adjusted *P* value < 0.01; ***, adjusted *P* value < 0.001. **e** Quantification of knockdown of *ZFP36* using independent siRNAs. Statistical significance was determined by ANOVA with Dunnett's multiple comparison test ($n = 3$). *, adjusted *P* value < 0.05; **, adjusted *P* value < 0.01. $n = 3$. **f** Effect of suppression of ZFP36/TTP on *NOXA* mRNA following Actinomycin D treatment ($n = 3$). Statistical significance was determined using extra sum-of-squares *F*-test. ***, adjusted *P* value < 0.001. **g** Effect of ZFP36 expression on *JUN*, *NOXA*, and *E2F4* mRNAs in A375M cells ($n = 3$ per group). Statistical comparison of ZFP36 expressing cells compared to control cells was done using *t* test. ***, adjusted *P* value < 0.001; **, adjusted *P* value < 0.01. **h** Quantification of *NOXA* mRNA associated with immunoprecipitated ZFP36. Statistical comparison of ZFP36 expressing cells compared to control cells was done using *t* test. ($n = 3$). **, adjusted *P* value < 0.01; ****, adjusted *P* value < 0.0001. **i** The sequence of ZFP36 with putative MAPK phosphorylation sites. **j** Effect of dabrafenib on wild-type and mutant ZFP36 in A375M cells. Error bars indicate mean ± s.e.m. of indicated replicates. See also Supplementary Fig. 3. Source data for all Western blots are provided as a Source Data file

tumors were palpable, then treated them with vemurafenib or dabrafenib by oral gavage. We measured anti-apoptotic dependence using DBP on extracted tumors 8–23 h after drug treatment. Overall apoptotic dependence to BRAF inhibition (BIM peptide) was observed within 8 h (Fig. 5b and Supplementary Fig. 5a). Similar to our in vitro data, dependence on MCL-1 (measured using the MS1 and NOXA peptides) was also induced following BRAF inhibition. BRAF inhibition in vivo also suppressed NOXA protein and promoted the association of BIM to MCL-1 (Supplementary Fig. 5b).

Finally, we tested the efficacy of sequential targeting of BRAF and MCL-1 in vivo. We treated mice bearing A375M tumors with the drug vehicle, the MCL-1 inhibitor S63845[19], dabrafenib alone, or dabrafenib followed 7 h later by the MCL-1 inhibitor (Fig. 5c). We observed no overt toxicity of any treatments or significant changes in weight during the treatment period (Fig. 5d). In the presence of dabrafenib, tumor growth was inhibited, but the growth inhibition was reversed upon discontinuation of treatment (Fig. 5e; Supplementary Fig. 5c), similar to prior results[49]. Although the MCL-1 inhibitor alone did not affect tumor growth, sequential BRAF and MCL-1 inhibition led to profound tumor regression. We validated these results using a structurally unrelated MCL-1 inhibitor currently in clinical trials, AZD5991[48]. Again, we observed that the sequential treatment leads to significant decreases in tumor volume compared with dabrafenib alone (Fig. 5e).

After 14 days, treatment in all arms was discontinued to evaluate the durability of the observed effects. Tumors in dabrafenib-treated animals resumed their growth, eventually requiring animal euthanasia (Fig. 5g). Treatment with BRAF inhibitors followed by MCL-1 inhibitors significantly prolonged survival compared with dabrafenib alone. Interestingly, we observed that many of these mice had small or undetectable tumors, even after > 30 days after treatment discontinuation (Supplementary Fig. 5c). To definitively establish the extent of tumor response, we excised the tumor site from several BRAF/MCL-1 inhibitor-treated mice. Among the animals evaluated, we observed that there was no pathologic evidence of residual tumors in three mice (Fig. 5h). Overall, these data suggest that MAPK inhibition can lead to dependence on MCL-1 in vivo and that therapeutic targeting using BRAF and MCL-1 inhibitors can induce profound tumor (in some cases pathologic complete responses) in tumor-bearing mice.

## Discussion

Small-molecule inhibitors of the MAPK signaling pathway, such as BRAF/MEK inhibitors, benefit the majority of patients whose tumors carry activating mutations in the target oncoproteins. However, complete responses to these drugs are infrequent, indicating that understanding the mechanisms of resistance to

MAPK pathway inhibitors could lead to improved effectiveness of therapy across many disease types. Multiple genetic and epigenetic mechanisms have been shown to contribute to BRAF/MEK inhibitor resistance, including mutations in alternative components of the MAPK pathway[50–53], activation of parallel signaling pathways such as PI3K/AKT[54], or expression of stromal cell factors[55,56]. These resistance mechanisms are typically detected after treatment of patients with the targeted agents, suggesting that the selective pressure of drug therapy reduces the dependence of the tumor on the oncoprotein being targeted.

In contrast to genetic or epigenetic resistance mechanisms, non-genetic adaptation to targeted drugs may also impact the therapeutic efficacy of targeted agents[57]. Although less well characterized, examples of adaptive resistance include the negative feedback pathways that are activated upon MAPK inhibition[35,58]. We have also shown that BRAF inhibitors induce a rapid adaptation in metabolic pathways related to the induction of the melanoma transcription factor *MITF*[37]. These adaptive responses can lead to tumor cells that survive initial phases of treatment, facilitating the eventual emergence of genetic subclones that are not dependent on the targeted oncoprotein.

In this study, we evaluated the mechanisms by which cancer cells evade apoptosis despite potent targeted suppression of their oncoprotein. Surprisingly, we find that adaptation, rather than genetic mechanisms, dampens the initial apoptotic response to MAPK inhibitors across multiple cell lineages. Specifically, we find that MAPK inhibitors lead to decreased levels of the pro-apoptotic factor *NOXA*. The suppression of NOXA correlates with a shift of MCL-1 from the weak pro-apoptotic activator NOXA[59] to BIM, increasing the dependence of the cell on MCL-1. Consistent with this mechanism, pre-treatment of melanoma or lung cancer cells with MAPK inhibitors, but not the alternative sequence, strongly sensitizes them to MCL-1 inhibitors. The shift of MCL-1 to the more potent activator BIM may account for the observed schedule-dependent cytotoxicity of MCL-1 inhibitors, consistent with the requirement of BIM for apoptosis to BRAF inhibitors[60]. Although our data suggest that MAPK suppression leads to adaptive changes in MCL-1 dependence, suppressed NOXA could also be a mechanism of intrinsic resistance. For example, Floros et al. have recently shown that reduced levels of the estrogen receptor ERα prevent the transcription of *NOXA*, leading to intrinsic resistance to HER2 inhibitors[61].

Mechanistically, our data suggest that the suppression of *NOXA* mRNA in response to targeted therapy are at least partially post-transcriptional. Prior reports have described several AU-rich decay elements in the NOXA 3′ untranslated region[50], consistent with our observations that ZFP36/TTP is essential for decay of NOXA in response to BRAF inhibitors. However, we cannot exclude the possibility of other post-transcriptional mechanisms (for example,

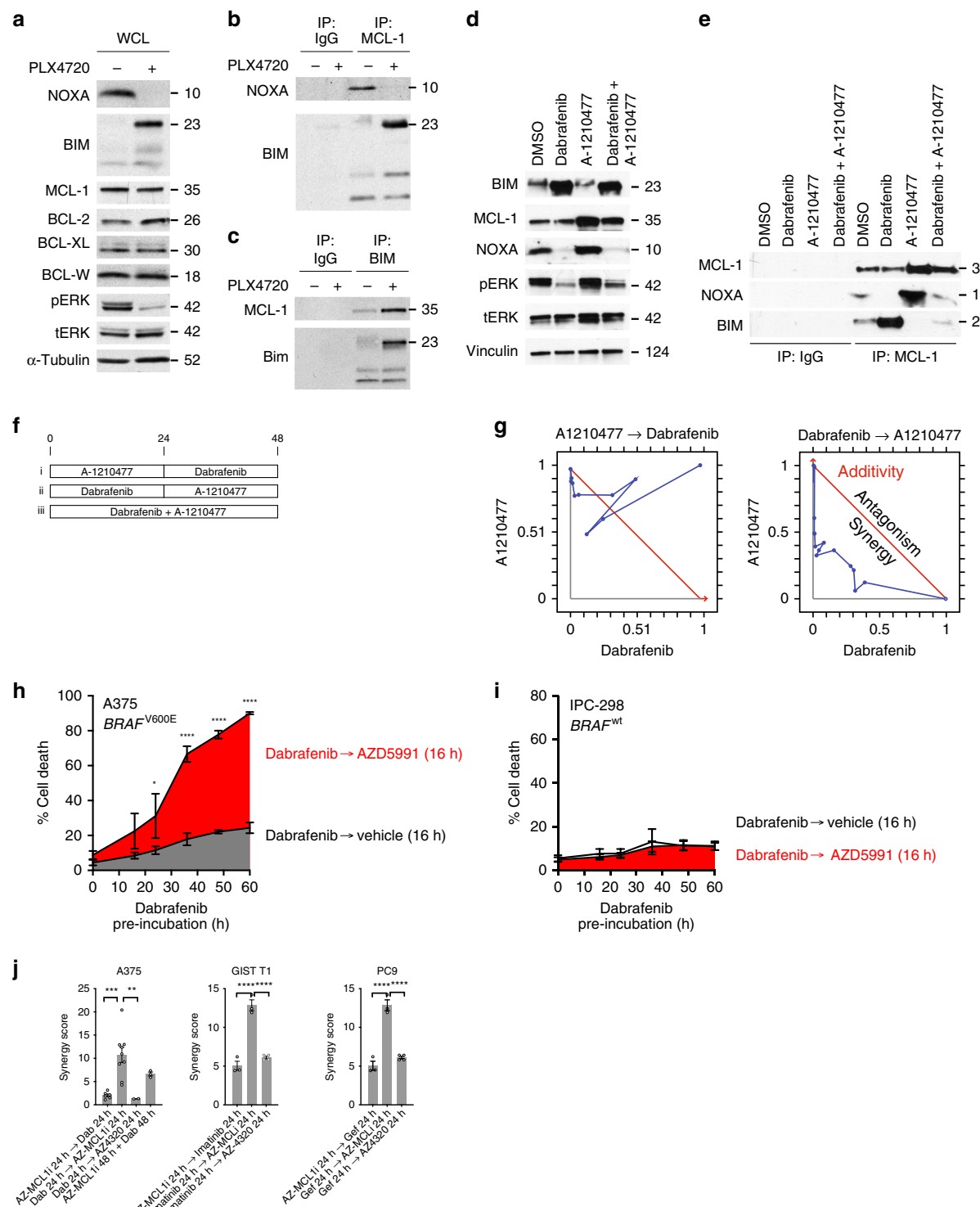

**Fig. 4** Targeted therapies reconfigure apoptotic signaling and create a schedule-dependent vulnerability. **a** Effect of PLX4720 (1 μM) on total levels of NOXA and BCL-2 family members in A375M melanoma cells (input for immunoprecipitation). **b** Association of MCL-1 with NOXA and BIM following PLX4720 treatment. **c** Association of BIM with MCL-1 following PLX4720 treatment. **d** Effect of A1210477, dabrafenib or both on MCL-1 and BIM. **e** Effect of A1210477, dabrafenib or both on the association of protein levels of MCL-1, BIM, and NOXA. **f** Scheduling of drug treatments in **g**. **g** Representative synergy blot of A-121044 followed by dabrafenib, or reverse, in A375M cells. **h** Effect of dabrafenib followed by AZD5991 on apoptosis in A375M cells. Cells were treated with vehicle (time 0) or dabrafenib (1 μM) for the indicated time, followed by 16 h exposure to AZD5991 (1 μM) or equal volume vehicle. Statistical comparison ($n = 4$) of different drug schedules was done using two-way ANOVA with Sidak's multiple comparison test. *, adjusted $P$ value < 0.05; ****, adjusted $P$ value, < 0.0001. **i** Effect of dabrafenib followed by A1210477 on apoptosis in IPC-298 cells ($n = 4$). Cells were treated as in **h**. **j** Synergy scores of A375M, GIST-T1 or PC9 cells treated with MCL-1 or BCL-2 inhibitors with targeted agents. Statistical comparison was done using one-way ANOVA with Holm–Sidak's multiple comparison test. ****, adjusted $P$ value < 0.0001; ***, adjusted $P$ value < 0.001; **, adjusted $P$ value < 0.01. Error bars indicate mean ± s.e.m. of indicated replicates. See also Supplementary Fig. 4. Source data for all western blots are provided as a Source Data file

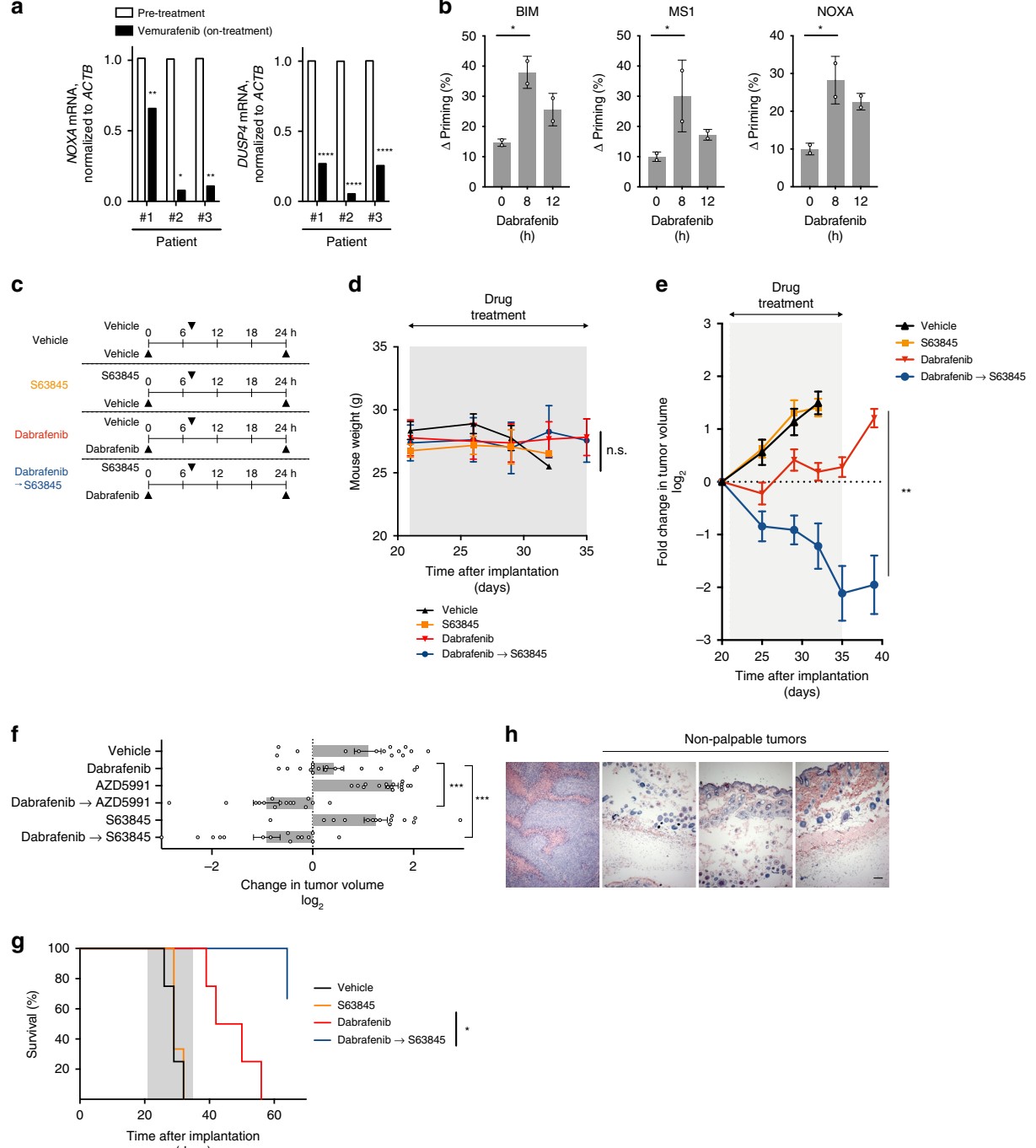

**Fig. 5** Targeting apoptotic adaptation overcomes resistance to targeted therapy in vivo. **a** Comparison of *NOXA* and *DUSP4* mRNA in paired biopsies obtained from melanoma patients before treatment with vemurafenib (pre-treatment) and 10–15 days later. Statistical comparison was done using ANOVA with Sidak's multiple comparison test. ****, adjusted *P* value < 0.0001; **, adjusted *P* value < 0.001; *, adjusted *P* value < 0.05. $n = 1$ with three technical replicates. **b** BH3 profiling of A375M melanoma xenografts ($n = 2$) following treatment with dabrafenib in vivo. BIM peptide measures BCL-2 family member dependence, whereas MS1 and NOXA peptides measure dependence on MCL-1. Statistical comparison was done using one-way ANOVA with Dunnett's multiple comparison test. *, adjusted *P* value < 0.05. **c** Timing of drug treatment of murine melanoma models. Arrows indicate time of drug treatment per day. **d** Weight of mice treated with BRAF inhibitor, MCL1 inhibitor S63845 or sequential administration of BRAF inhibitor followed by MCL1 inhibitors ($n = 8$ per group). **e** Change in A375M xenograft tumor volume following treatment with MCL1, BRAF, or sequential BRAF/MCL1 inhibitors. Statistical comparison of change of dabrafenib versus dabrafenib → S63845-treated animals was done using two-away ANOVA with Sidak multiple comparison tests ($n = 14$–16). **, adjusted *P* value < 0.005. **f** Change in A375M xenograft tumor volume after 14 days treatment (relative to pre-treatment tumor volume) following treatment with structurally distinct MCL1 inhibitors, BRAF inhibitor, or sequential administration of BRAF inhibitor followed by MCL1 inhibitors. Statistical comparison with vehicle-treated animal was done one-way ANOVA with multiple comparison tests. ***, adjusted *P* value < 0.001. **g** Overall survival of A375M xenograft models following treatment with MCL1, BRAF or sequential BRAF/MCL1 inhibitors. Statistical comparison was performed using Log-rank (Mantel–Cox) test. *, *P* value < 0.05. **h** Evaluation of residual tumors in mice treated with dabrafenib followed by S63845. Scale bar = 100 μm. For **d**, **e**, **f**, $n = 14$–16 per group. Error bars indicate mean ± s.e.m. of indicated replicates. See also Supplementary Fig. 5. Source data for all Western blots are provided as a Source Data file

other AU-rich binding proteins) or other gene regulatory mechanisms such as transcription may also play roles in NOXA expression. As the ZFP36 family includes three AU-binding proteins, further work will need to be performed to clarify the generality of ZFP36 essentiality in NOXA suppression.

Our findings have broad implications for the rational use of anti-apoptotic inhibitors for solid tumor patients. Existing data from lung cancers models[33] and limited solid tumor cell lines[19,62] suggest that manipulating the MCL-1/NOXA axis may enhance the efficacy of MAPK pathway inhibitors. However, our data indicate that the timing of MCL-1 inhibitors will need to be carefully considered in relation to MAPK inhibitors, given the adaptation mechanism described here. We found that pre-treatment with MAPK inhibitors sensitized to subsequent MCL-1 inhibition, whereas the converse treatment schedule exhibited limited synergy. We found that concomitant treatment of melanoma cells with inhibitors of BRAF and MCL-1 was also less effective than sequential treatment (Fig. 4j). Thus, sequencing might not only increase the efficacy of drug therapy but has the potential to improve the tolerability of combination therapy, given the respective toxicities of each drug. The timing of these drug treatments in clinical settings will require challenging optimization of schedule, owing to the specific pharmacokinetic and pharmacodynamic characteristics of the drugs. The kinetics of NOXA loss might provide some information about optimal sequencing strategies, however, NOXA levels decrease rapidly at both mRNA (Fig. 3a) and protein (Fig. 2a), yet the MCL-1 dependence follows slightly thereafter (Supplementary Fig. 1d). Further, optimal timing strategy will require additional measurement of BIM levels, given that MCL-1 dependence following MAPK inhibitor treatment reflects increases in BIM as well as suppression of NOXA. Owing to these multiple effects, we hypothesize that a superior tool to evaluate optimal dosing is BH3 profiling, which we performed in our pre-clinical model (Fig. 5b) and be extended to clinical settings.

An equally important factor in the translation of these findings is the selection of patients most likely to benefit from the described MAPKi/MCL-1 inhibitor strategy. As apoptotic adaptation requires the MAPK-mediated suppression of NOXA, we anticipate that combined MAPKi/MCL-1i therapy will not be effective in cancers where MAPKi resistance is associated with the failure to suppress to the MAPK pathway. Consistent with this hypothesis, we found that MCL-1 dependence was minimally changed after BRAF inhibition in melanoma cells derived from a patient who was resistant to BRAF/MEK inhibitors (Fig. 1f). Thus, targeting adaptive resistance may be most useful in patients before the emergence of MAPKi resistance.

Given that resistance of tumor cells to cell death is a hallmark of cancer, drugs that directly target apoptotic pathways have been the focus of intense pharmaceutical interest[63–65]. For example, several authors have previously shown that the addition of ABT-737, an inhibitor to anti-apoptotic BCL-2 and BCL-XL, partially sensitized melanoma cells to BRAF inhibitors[66,67]. In xenograft models, navitoclax enhanced the efficacy of BRAF inhibitors[49,68], leading to clinical trials combining this drug with BRAF/MEK inhibitors in melanoma patients (e.g., clinical trial NCT01989585). Given our findings, the translation of anti-apoptotic therapies may be aided by consideration of the dynamic changes in apoptotic signaling induced by drug treatment. We suggest that the use of DBP in clinical settings could enable the correct use of these BH3 mimetics to specific populations that would be likely to respond to treatment.

## Methods

**Cell lines and tissue culture**. Cancer cell lines were obtained from the Center for Molecular Therapeutics (CMT), based at Massachusetts General Hospital, were cultured in DMEM, RPMI-1640, or 1:1 mixture of DMEM:Ham's F12 media supplemented with 10% fetal bovine serum (FBS) and penicillin–streptomycin–glutamine. Human primary melanocytes were produced from discarded foreskin were grown in Ham's F10 media supplemented with 7% FBS, 1% PSQ, 100 μM IBMxz, 1 mM dbcAMP, 50 ng/mL TPA, and 1 μM sodium vanadate[36]. All cells were incubated at 37 °C in 5% $CO_2$.

**Clinical samples**. Patients with *BRAF*-mutant melanoma, who were not part of a clinical trial, were treated with vemurafenib for 10–15 days. Tumors were biopsied before treatment and at the end of treatment. All patients gave informed consent for tissue acquisition as per IRB-approved protocol 05–042 or 11–181 (Office for Human Research Studies, Dana-Farber/Harvard Cancer Center). Samples were previously utilized[36].

**Bioinformatics and statistics**. Effects of targeted therapies on NOXA mRNA was extracted from GSE51115, GSE1922, GSE19567, GSE57156, GSE34228, GSE50803, and GSE6184). Levels of NOXA were normalized relative to control treatment.

Measurement and statistics were made from distinct samples unless otherwise indicated in the legend figure.

**Buffers and reagents**. Lysis buffer contained 150 mM NaCl, 1.0% Triton X-100, and 50 mM Tris (pH 8). CHAPS immunoprecipitation buffer contained 0.5% CHAPS in HEPES buffer. PLX4720 was obtained from Sai Advantium Pharma Limited (Pune, India). PD0325901 was obtained from Santa Cruz Biotechnology (Dallas, TX). Gefitinib, crizotinib, imatinib, and lapatinib were obtained from LC Laboratories (Woburn, MA). SB590885, vemurafenib were obtained from Selleck Chemicals. S63845 was obtained from Chemgood (Glen Allen, VA). PI-103 was kindly donated by J. Engelman (Massachusetts General Hospital). Actinomycin D was obtained from Sigma-Aldrich (St. Louis, MO). A1210477 was obtained from Active Biochem (Hong Kong). AZD4320 and AZD5991 were kindly provided by Astra-Zeneca (Waltham, MA). Unless indicated, dose of the targeted therapies used was 1 μM for 24 h, except for trametinib which was used at 100 nM.

**Antibodies**. The following primary antibodies were used for western blot at 1:1000 dilution, unless otherwise noted: mouse anti-NOXA (Enzo Life Sciences, ALX-804–408), rabbit anti-Bcl-2 (Cell Signaling Technology, #2870 and #15071), rabbit anti-Bcl-w (CST, #2724), rabbit anti-Bcl-xl (BD Biosciences, #610211), rabbit anti-Mcl-1 (CST, #4572), rabbit anti-Bim (CST, #2933), rabbit anti-phospho-EGFR (Y1068) (CST, #2234), rabbit anti-EGFR (CST, #4267), rabbit anti-phospho-HER2 (Y1248) (CST, #2243), rabbit anti-HER2 (CST, #2165), rabbit anti-phospho-c-KIT Y719 (CST, #3391), rabbit anti-c-KIT (CST, #3074), rabbit anti-phospho-ALK (CST, #3341), rabbit anti-ALK (CST, #3633), rabbit anti-phospho-MET Y1349 (CST, #3121), rabbit anti-MET (CST, #8198), rabbit anti-phospho-ERK (CST, #9101), rabbit anti-ERK (CST, #4695), mouse anti-alpha- tubulin (Sigma-Aldrich, T9026, 1:5000), vinculin (Abcam ab129002), Cleaved PARP CST #5625). The following HRP-linked secondary antibodies were used: horse anti-mouse-IgG (CST, #7076), goat anti-rabbit-IgG (CST, #7074), and mouse anti-rabbit-IgG (conformation specific) (CST, #5127). Anti-V5 antibody or affinity gel was obtained from Thermo-Fisher Scientific. The following antibodies were used for immunoprecipitation: mouse anti-Mcl-1 (BD, #559027), rabbit anti-Bim (CST, #2933), normal mouse IgG (Santa Cruz, #2025), normal rabbit IgG (Santa Cruz, #2027), Protein A magnetic beads (CST, #8687), Protein G magnetic beads (CST, #8740).

**siRNAs**. siRNA pools (25 nM) targeting individual BCL-2 family members were obtained from Dharmacon. The following siRNA pools were used: siControl: SMARTpool D-001810-10-05; si*BCL2*: SMARTpool L-003307-00-0005; si*BCL2L1*: SMARTpool L-003458-00-0005; si*BCL2L2*: SMARTpool L-004384-00-0005; si*MCL1*: SMARTpool L-004501-00-0005; si*BCL2A1*: SMARTpool L-003306-00-0005; si*BAK1*: SMARTpool L-003305-00-0005; si*BAD*: SMARTpool L-003870-00-0005; si*BID*: SMARTpool L-004387-00-0005; si*HRK*: SMARTpool L-008216-00-0005; si*BBC3*: SMARTpool L-004380-00-0005; si*NOXA*: SMARTpool L-005275-00-0005; si*BCL2L11*: SMARTpool L-004383-00-0005; si*BECN1*: SMARTpool L-010552-00-0005; si*BIK*: SMARTpool L-004388-00-0005.

For experiments targeting ERK1/2, control siRNAs (Cell Signaling #6560 S) or siRNAs targeting *ERK1/2* Cell Signaling Technology (#6568 S) were used.

For experiments targeting ZFP36 family members, the following pooled siRNAs were obtained from Dharmacon: si*ZFP36*, SMARTpool L-010789-01; si*ZFP36L1*, SMARTpool L-011816-00-0005; si*ZFP36L2*, SMARTpool L-013605-01-005. Where individual knockdown of ZFP36 was required, siRNAs were obtained from Sigma-Aldrich: #1, WD02906089-004; #2, WD02906085; #3, WD02906087.

**Western blot**. Whole-cell lysates were collected in lysis buffer supplemented with cOmplete ULTRA protease inhibitor (Roche) and Phospho-STOP phosphatase inhibitor (Roche). After protein quantities were normalized using BCA Protein Assay Kit (Thermo Scientific), samples were denatured with SDS loading dye at 95 °C for 5 min. When probing for NOXA, samples were resolved on a 15% polyacrylamide gel at 35 mA for 25 min. For all other proteins, samples were

resolved on 4–15% or 10–20% TGX Criterion gradient gels (Bio-Rad) at 200 V for 35 min. Proteins were transferred to 0.2 μm nitrocellulose membrane at 100 V for 20 min (for NOXA) or 50 min (for all other proteins). The membrane was blocked with 5% milk in TBST for 45 min, washed with TBST, and incubated with primary antibodies at 4 °C for three nights (for NOXA) or overnight (for all other proteins). After washing with TBST, the membrane was incubated with secondary antibody at room temperature for one hour. Membranes were washed with TBST, and chemiluminescence reaction was performed using ECL Western Blotting Substrate (Pierce). Chemiluminescent films were exposed to the membrane in a darkroom and developed using Kodak X-OMAT 2000A.

**Immunoprecipitation.** Whole cell lysates were collected in CHAPS immunoprecipitation buffer supplemented with cOmplete ULTRA protease inhibitor (Roche) and Phospho-STOP phosphatase inhibitor (Roche). Protein concentration was measured using BCA Protein Assay Kit (Thermos). For each sample, 750 μg protein was aliquoted into two 1.5 mL Eppendorf tubes. Antibodies for the protein of interest and control IgG were incubated with the samples overnight at 4 °C on a rocker. Protein A or Protein G magnetic beads, depending on the species of the primary antibody, were added, and the samples were rocked for an additional 45 min at 4 °C on the rocker. The beads were precipitated three times using a magnetic stand apparatus and washed with fresh CHAPS buffer after each precipitation. After the third wash, SDS loading dye was added, and samples were boiled at 95 °C for 5 min. Samples were resolved by western blot.

**Quantitative real time-polymerase chain reaction (RT-PCR).** Total RNA was collected and purified using QiaShredder and RNeasy Plus Mini Kit as indicated (Qiagen). RT-PCR was performed using Universal Fast SYBR kit (Kapa Biopsystems), and amplification was measured with ABI Real Time PCR System. Expression levels were normalized to *ACTB*. The following primers were used (listed 5′–3′):

*PMAIP1*: Forward: AAGTTTCTGCCGGAAGTTCA. Reverse: GCAAGAACGCTCAACCGAG

*TRPM1*: Forward: CAAAGATACATTCCCGTTTGC. Reverse: GCTGAAAGAGCCTGAGCTGT

*DUSP4*: Forward: CCCACAGAGCAGTATTAGGCTGAAG. Reverse: CAGCGTGGATGAGCAACTGAA

*ACTB*: Forward: GTTGTCGACGACGAGCG. Reverse: GCACAGAGCCTCGCCTT

*CCND1*: Forward, CACACGGACTACAGGGGAGT. Reverse, CACAGGAGCTGGTGTTCCAT

*CCND1*: Forward, CCA AAG GAT AGT GCG ATG TTT. Reverse, CTG TCC CTC TCC ACT GCA AC

**RNA immunoprecipitation.** RNA immunoprecipation protocol was adapted from prior studies[69]. Cells were grown until 90% confluent before being washed twice with PBS. Cells were then scraped from the plate and collected by centrifugation at 3000 *g* for 5 min 4 °C. The cells were then lysed (10 mM HEPES, 100 mM Potassium Chloride, 5 mM Magnesium Chloride, 25 nM EDTA, 0.5% IGEPAL, 2 mM DTT, 0.2 mg/ml Heparin, Protease inhibitor and RNase inhibitors) and the lysate centrifuged to 10 mins at 4 °C. The supernatant was collected and quantitated prior to freezing aliquots in liquid nitrogen for the RNA immunoprecipitation (RIP) and the concentration determined. Equal amounts of protein from control (V5-empty) or V5-ZFP36-expressing cells were incubated with either anti-V5 Sepharose beads or IgG control. The RNA immunoprecipitation was incubated overnight with constant agitation at 4 °C. To purify the V5-ZFP36, the beads were pelleted by centrifugation at 10,000 rpm for 3 mins and then washed three times with 1 ml of wash buffer (50 mM Tris, 150 mM Sodium Chloride, 1 mM Magnesium Chloride, 0.05% IGEPAL and 0.02 mg/ml Heparin). The purified control or V5-ZFP36 was removed from the beads using SDS-sample buffer lacking bromophenol blue and the bound protein analyzed using western blot.

To identify the RNA bound in control and V5-ZFP36 lysates from both IgG and V5 RIPs, we purified the RNA from the SDS-sample buffer RIP elution using Qiagen RNeasy columns and protocol. Equal amounts of purified RNA from each RIP experiment was used to make cDNA using the ROCHE cDNA synthesis kit (Cat# 04379012001) as per the manufacturers' instructions. RT-PCR experiments conducted using primers detailed in oligo section of methods.

**siRNA sensitization screen and overexpression.** Cell lines were counted and plated in 96-well dishes[36]. siRNA transfections were conducted at the time of plating of cells using the lipidoid delivery agent C12-133-B[15]. Each experiment was conducted in triplicate. Cell number was quantified at 72 h after transfection/ plating using Cell Titer Glo assay (Promega). Data were normalized to cells transfected with control siRNA treated with drug vehicle. Validation of siRNA screen results was done using individual siRNAs targeting the 3′ untranslated region of *MCL1* (Dharmacon).

pLv-105 Noxa lentiviral plasmid was obtained from Genecopoeia (Rockville, MD). ZFP36 cDNA was obtained from Harvard's PlasmID database and cloned using Clonase LR into pLX304 DEST. Site-directed mutagenesis was done using In-Fusion

HD (Clontech). Successful mutagenesis was confirmed by Sanger sequencing. Lentivirus was produced using Lenti-X 293 T cells using standard methods.

**Microarray analysis.** Microarray data were retrieved from Microarray Gene Expression Omnibus (http://www.ncbi.nlm.nih.gov/geo). Accession numbers GSE10086[34] and GSE20051[70] were analyzed for changes in BCL2 family mRNA using GraphPad Prism. NOXA levels in melanoma cells were extracted from GSE51115.

**Validation of NOXA suppression.** SK-MEL-5 ($1 \times 10^5$ mL$^{-1}$), WM1575 ($1 \times 10^5$ mL$^{-1}$), MALME ($2 \times 10^5$ mL$^{-1}$), A375M ($1 \times 10^5$ mL$^{-1}$), and human primary melanocytes ($3.5 \times 10^5$ mL$^{-1}$) were each plated in 10 cm plates and 24-well plates. Cells were treated with 1 μM PLX4720, 10 nM PD0325901, or DMSO for 24 h. Protein and RNA lysates were collected as described above from the 10 cm plates and 24-well plates, respectively. Protein expression of NOXA, phospho-ERK, ERK, and alpha-tubulin was measured by western blot. The mRNA expression of *NOXA* and *ACTB* was measured by quantitative RT-PCR. Similar experiments were run for RTK-activated cells. All cell lines (GTL16, EBC1, SH-SY5Y, KELLY, WIDR, GIST882, BT474, CALU1) were plated at $1 \times 10^5$ cells mL$^{-1}$ in 10 cm plates and 24-well plates. GTL16, EBC1, and CALU1 cells were treated with either 1 μM gefitinib or DMSO. SH-SY5Y and KELLY cells were treated with either 1 μM critzotinib or DMSO. GIST882 cells was treated with either 1 μM imatinib or DMSO. BT474 cells were treated with either 1 μM lapatinib or DMSO. Protein expression of Noxa, phospho-RTK, RTK, phospho-ERK, ERK, and alpha-tubulin were measured by western blot. The mRNA expression of *NOXA* and *ACTB* was measured by quantitative RT-PCR.

**Time course experiment of NOXA suppression.** SK-MEL-5 ($1 \times 10^5$ mL$^{-1}$), MALME ($2 \times 10^5$ mL$^{-1}$), and A375M cells ($1 \times 10^5$ mL$^{-1}$) were plated in 24-well plates. MALME cells were also plated in 10 cm plates. Cells were treated with 1 μM PLX4720 at six time intervals (0, 1, 2, 4, 8, 24 h). Total RNA lysates from all samples were collected at the same time, and quantitative RT-PCR for *NOXA*, *TRPM1*, and *ACTB* was performed using the aforementioned protocol. Protein lysates were collected from MALME cells and were probed for Noxa, phospho-ERK, ERK, and alpha-tubulin by Western blot.

**Actinomycin D assay.** MALME melanoma cells ($1.25 \times 10^5$ mL$^{-1}$) were plated in 24-well plates. After overnight incubation at 37 °C, Actinomycin D and PLX4720 were added at six time intervals (0, 1, 2, 4, 8, 24 h). At the start of each interval, cells were treated with drugs with one of three ways: (1) 30 min pre-incubation with 10 μg/mL Actinomycin D before addition of DMSO, (2) 30 min pre-incubation with 10 μg/mL Actinomycin D before addition of 1 μM PLX4720, or (3) 30 min pre-incubation with DMSO before addition of 1 μM PLX4720. Total RNA lysates from all samples were collected at the same time and quantitative RT-PCR for *NOXA*, *TRPM1*, and *ACTB* was performed using the aforementioned protocol. Quantification of transcript abundance was compared to time at which Actinomycin D was added ($t = 0$).

**Cell death assays.** Cells were treated as indicated and stained with fluorescent conjugates of annexin-V and PI (1 μg/ml final concentration) and analyzed on a FACSCanto machine (BD, Franklin Lakes, NJ, USA). Annexin-V was prepared are previously described[71]. Viable cells are Annexin-V negative and PI negative, and cell death is expressed as 100%−viable cells.

**Dynamic BH3 Profiling.** DBP was performed as previously described in detail[29]. Cell lines or primary melanoma cells were incubated ($1.5–3 \times 10^5$/ml) in RPMI with 10% FBS at different times and drug concentration as indicated. To perform DBP in cell lines, $2 \times 10^4$ cells/well were used for. In total, 15 μL of BIM BH3 peptide (final concentration of 0.03, 0.1, 0.3, 1, and 3 μM), BAD BH3 peptide (10, 100 μM), NOXA BH3 peptide (100 μM), MS1 BH3 peptide (10) or HRK BH3 peptide (100 μM) in MEB (150 mM Mannitol, 10 mM Hepes-KOH pH 7.5, 50 mM KCl, 0.02 mM EGTA, 0.02 mM EDTA, 0.1% BSA, 5 mM succinate) were deposited per well in a black 384-well plate (BD Falcon no. 353285). Single cell suspensions were washed in MEB before being resuspended at $4 \times$ their final density. One volume of the $4 \times$ cell suspension was added to one volume of a $4 \times$ dye solution containing 4 μM JC-1, 40 μg/mL oligomycin, 0.02% digitonin, 20 mM 2-mercaptoethanol in MEB. This $2 \times$ cell/dye solution stood at RT for 10 min to allow permeabilization and dye equilibration. A total of 15 μL of the $2 \times$ cell/dye mix was then added to each treatment well of the plate, shaken for 15 s inside the reader, and the fluorescence at 590 nm monitored every 5 min at RT. Percentage loss of Ψμ for the peptides is calculated by normalization to the solvent only control DMSO (0% depolarization) and the positive control FCCP (100% depolarization). Individual DBP analysis were performed using triplicates for DMSO, FCCP, and the different BH3 peptides used, and the expressed values stand for the average of three different readings. In cases were standard deviation was > 10%, the outlying reading was discarded. % priming stands for the maximum % depolarization obtained from the different BH3 concentrations tested. Δ% priming stands for the difference between treated cells minus non-treated cells (% priming$^{treated}$ −% priming$^{non-treated}$).

We used a FACS-based BH3 profiling to perform the analysis, cytochrome c release was measured after a 60 min incubation of digitonin-permeabilized cells with BH3 peptides, as previously described[32]. Antibodies used were: Zombie Aqua Dye (Biolegend, #423101), anti-hFAP (PE conjugated, R&D systems #FAB3715P), CD45 (BD Horizon, BV421clone HI30), and anti-hNG2/MCSP (R&D systems, #FAB2585F, FITC) and cytochrome c-Alexa Fluor 647 (Biolegend, #612310).

**Patient primary cell isolation.** Primary metastatic melanoma tumors were exposed to an enzymatic digestion after, mechanical disgregation, in 2.5 mL of DMEM/F12 media with 125 units of DNAse I (Sigma-Aldrich #DN25), 100 units of Hyaluronidase (Sigma-Aldrich #H3506) and 300 units of collagenase IV (Gibco #17104–019). The tissue suspension was processed using gentleMACS Dissociator (Miltenyi Biotec) using the hTUMOR 1 program. The suspension was incubated at 37°C for 30 min in constant agitation. After the program hTUMOR 1 was ran again and repeated the 30 min incubation. We filtered the suspension 70 micron filter into a 50 mL conical and cells were spinned down $400 \times g$ for 5 min. To lyse the residual red blood cells, 100 μL of ice cold water was added for 15 s and then diluted to 50 mL with PBS, then spin cells down again. Cells were finally resuspended in RPMI media.

**Effect of sequential treatment of cells with BRAF and Mcl-1 inhibitors.** Cells were plated at $1–2.5 \times 10^5$ cells mL$^{-1}$. Twenty-four hours later, serial diluted drug was added to cells. After 24 h, media was removed and replaced with fresh, pre-warmed media with the second drug. Cell number was estimated after 24 h additional hours using Cell Titer Glo. Synergy scores were calculated using the Chalice Analyzer (Horizon Discovery).

**Animal experiments.** In total, $6.5 \times 10^6$ A375M cells were subcutaneously injected into both flanks of 6-week-old male Nu/Nu mice (Charles River Laboratories, #088). Three weeks after implantation, mice were randomized into four to six groups and treated daily up to 14 days with either vehicle or dabrafenib 30 mg/kg by oral gavage, followed 7 h later by either vehicle, AZD5991 100 mg/kg, or S63845 25 mg/kg by intravenous injection. Mouse weight and tumor volume were monitored twice a week. Mice were killed when tumor volume reached 1300 mm$^3$ and overall survival was monitored. All experiments were performed in compliance with federal laws and institutional guidelines and were approved by the Animal Care and Use Committee of the Dana-Farber Cancer Institute.

**Immunohistochemistry.** Tumors were harvested, fixed overnight in formalin 10%, and stored in ethanol 70%. Samples were submitted to the Brigham and Women's Hospital Pathology Core for paraffin embedding, sectioning, and hematoxylin and eosin staining. Imaging was performed using the Nikon Eclipse Ti-E.

**Reporting summary.** Further information on research design is available in the Nature Research Reporting Summary linked to this article.

## Data availability
siRNA screen data (Fig. 1b) are available in the Source Data file. Raw, uncropped western blots (Figs. 2a–f, 3j, 4a–e, and Supplementary Figs. 1b–d, 2a–2l, 3d, e, i, 4a–e, 5b) are provided as a Source Data file. All relevant data can be inquired from the corresponding authors.

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

## Acknowledgements

R.H. acknowledges funding from Melanoma Research Alliance and from the O'Connor-MacGregor Fund for Melanoma Research. J.M. and A.L. acknowledge support SU2C and The V Foundation (TVF) SU2C-TVF Convergence Scholar Awards (Grant # Grant #D2015-037). J.M. acknowledges Ramon y Cajal Programme, Ministerio de Economia y Competitividad (RYC-2015–18357). D.E.F acknowledges support from the National Institutes of Health (5P01 CA163222, 1R01CA222871, 5R01AR072304 and 5R01 AR043369), the Melanoma Research Alliance, and the Dr. Miriam and Sheldon G. Adelson Medical Research Foundation.

## Author contributions

J.M. conducted the BH3 profiling experiments and developed the BH3 profiling of primary melanoma samples. R.H., D.K., C.G., and D.S. performed western blots, immunoprecipitations, quantitative PCR, and actinomycin D experiments. C.G. performed the in vivo experiments. W.M. performed RNA immunoprecipitation. F.S.H., M.M, K.F., and C.Y. provided clinical samples from melanoma patients. J.R.C., J.P.S., and A.E.T. provided anti-apoptotic family inhibitors. R.H., D.E.F., J.M., and A.L. interpreted the data and wrote the manuscript with contributions from all authors. R.H., J.M., and D.E.F. conceived the study, designed the research, and planned the experiments.

## Competing interests

K.F. is a member of the board of directors at Loxo Oncology, Clovis Oncology, Strata Oncology, Vivid Biosciences; the corporate advisory board of X4 Pharmaceuticals, PIC Therapeutics; the scientific advisory board of Sanofi, Amgen, Asana, Adaptimmune, Fount, Aeglea, Array BioPharma, Shattuck Labs, Arch Oncology, Tolero, Apricity, Oncoceutics, Fog Pharma, Tvardi; and consultant for Novartis, Genentech, Bristol-Myers Squibb, Merck, Takeda, Verastem, Checkmate, Boston Biomedical. F.S.H. is a consultant for Bristol-Myers Squibb, Merck, EMD Serono, Novartis, Celldex, Amgen, Genetech, Incyte, Bayer, Aduro, Partners Therapeutics, Sanofi, Pfizer, Pionyr; member of scientific advisory board of Apricity; member of the advisory board of Pionyr, 7 Hills Pharma, Verastem. R.H. has received research grants from Bristol-Myers-Squibb and Novartis. J. R.C., A.E.T., and J.P.S. are employees of Astra-Zeneca. C.Y. is a consultant to Merck. J.M. is a consultant for Vivid Biosciences and Oncoheroes Biosciences. D.E.F. has a financial interest in Soltego, Inc., a company developing SIK inhibitors for topical skin darkening treatments that might be used for a broad set of human applications. These interests were reviewed and are managed by Massachusetts General Hospital and Partners HealthCare in accordance with their conflict of interest policies. The remaining authors declare no competing interests.
