## [Peer Review File · Nature Communications]

Reviewers' comments:

Reviewer #1 (Remarks to the Author):

In this well-conducted study, Haq and colleagues define NOXA mRNA destabilization as a common mechanism of adaptive resistance to targeted therapies. The notion that cell death induction by targeted therapies can be accentuated by treatment with BH3 mimetics is now quite established. However, even in this context, the current study offers several important insights that I believe will propel the field forward. Among these, the authors nicely demonstrate that MCL-1 is likely to be the best BCL-2 family member to target in the context of targeted therapy treatment. Technically, the work is very impressive, making use of sophisticated dynamic BH3 profiling assays conducted in a flow-based format with cell type-specific sorting, dynamic protein immunoprecipitation experiments (which are sometimes challenging to perform with BCL-2 family proteins), and beautiful *in vivo* studies with drug scheduling. Further, they articulate a mechanism by which NOXA mRNA levels are controlled by ERK-mediated phosphorylation of ZFP36. In light of these strengths, I have only minor questions and suggestions, as follows:

1. The authors nicely show that in BRAF mutant melanoma cells, NOXA decay following RAFi treatment is ZFP36-dependent, and it is likely regulated specifically by ERK-mediated ZFP36 phosphorylation at sites S218 and S228. It would be nice, if feasible, to (1) show that ZFP36 is differentially phosphorylated in the presence of MAPKi; (2) define whether ZFP36 is a direct ERK1/2 substrate; and (3) determine whether ZFP36 is required for drug-induced NOXA mRNA loss (as in Figure 3F) in other targeted therapy contexts (e.g., EGFR-, MET-, c-KIT, ALK-, and/or HER2-driven cells treated with their cognate targeted therapies).
2. The work on scheduling MAPKi  MCL-1i is very nice, but a few opportunities to further clarify the authors thinking should be considered. First, beyond the obvious notion that two drugs are often more toxic than one, why do the authors pursue scheduling without much attention toward simply administering the drugs in parallel? Second, in the *in vivo* scheduling experiments (where results are quite impressive), how exactly did the scheduling work? I found it difficult to understand exactly how the mice were dosed in these experiments. Finally, one can imagine that knowledge of the kinetics of drug-induced NOXA loss, and subsequent NOXA rebound following drug withdrawal, could directly inform the kinetics of optimal scheduling. Can the authors discuss this in more detail?
3. Small point: In Discussion, Line 43, "rationale" should be spelled "rational"

Finally, the statistical analyses used in this study appear to be appropriate.

Reviewer:
Kris Wood
Duke University

Reviewer #2 (Remarks to the Author):

Experiments described in this manuscript present an innovative and compelling demonstration that oncogene-targeted therapeutic approaches induce apoptotic resistance through the anti-apoptotic protein MCL-1. This is a question of profound significance, given the inability of most targeted cancer therapies to produce sustained responses. The authors show that therapy-induced resistance is accompanied by suppression of the pro-apoptotic MCL-1-inhibitory factor NOXA in an Erk-dependent manner, at least in part via destabilization of NOXA mRNA via the RNA-binding protein ZFP36. NOXA competes for binding to MCL-1 with the anti-apoptotic factor Bim, suggesting that inhibitors of the MCL-1:Bim interaction might prove useful adjuvants to current oncogene-targeted therapies, a model supported by data from both cultured cell and tumor xenograft model

systems.

Much of the study is admirably rigorous, using orthogonal approaches in multiple established cell lines, primary tumor cells, and murine xenografts for several key experiments. The impact of the work is also broadly applicable, as this acquired apoptotic resistance mechanism was observed across a range of drugs targeting different driver oncoproteins. Critical RNAi experiments included multiple independently-targeted siRNAs to ensure response specificity. Loss- and gain-of-function strategies are frequently employed, with complementary data from both in some circumstances. With a subset of experiments, however, there are concerns in presentation and/or interpretation:

1. In Figs. 3B, 3F, and Suppl. Fig. S3B, mRNA decay data should be plotted as a function of actinomycin D treatment, since this is the reagent that silences transcription (also, for one data set in each panel, actinomycin D is the only reagent added). The description of these experiments in the Methods section is confusing and does not reflect the data presented. This is especially true for the time points listed as well as the actinomycin dose given (10 mg/ml), which is not soluble at that concentration in aqueous solution.

2. Based on the data provided in Fig. 3, the contribution of accelerated mRNA decay to suppression of NOXA mRNA (and protein) levels appears to be real but minor. In Fig. 3A (and Fig. 2E), treatment with BRAF or MEK inhibitors decreased NOXA mRNA levels by 75-80% (it is also noted that the magnitude of the decrease in NOXA protein was not calculated), yet Fig. 3B only shows a 30% or so acceleration of mRNA decay. It is highly probable that other gene regulatory mechanisms, particularly transcription, play major roles in repressing NOXA expression in this system. If ZFP36-directed mRNA decay is the major mechanism accounting for suppressing NOXA following treatment with targeted therapies, then NOXA levels should be unaffected by such treatments in ZFP36-silenced cell models; however, this experiment was not reported. In the absence of this experiment, these points should be addressed in the Discussion as a likely limitation of the proposed mechanism for NOXA suppression.

3. The standard for validation of siRNA efficacy is to demonstrate protein suppression, as the authors do in Suppl. Fig. S2L, since validating by target mRNA level alone often overestimates knockdown efficiency for stable proteins while also ignoring potential effects of translational suppression. This concern applies to Figures S1A, S3C, and 3E.

4. Statistical significance is not indicated in many figure panels, even though often indicated in the text. For example, no indication of significance is included for results of BIM treatment in Fig. 1C (or anywhere in 1E, 1F). Other examples include Figs. S1F, S2M, 3D, and 3F.

5. In Supplemental Fig. S5B, close cropping of the Western blots for MCL-1 is partially obscuring a higher MW band appearing in both the IP and WCL lanes, particularly for drug treated tumor #1 (and possibly vehicle-treated tumor #3?). The authors should show and explain (or speculate) on the appearance of this band, particularly since it does not seem to appear in any of their other blots using the MCL-1 antibody.

Minor points:

1. The language and grammar of select sections of the manuscript are difficult to follow, particularly in the Discussion where a few sentences make no sense as written.
2. A wide variety of inhibitors are used in this study. A supplemental table listing drugs used along with their targets would be helpful for non-experts in field.
3. p.4 line 25: inhibitor for ERBB2 lines left out of list
4. p.4 lines 44-47: "suppression of all BCL-2 family members (using the BIM peptide) induced a

significant increase in overall mitochondrial outer membrane permeabilization (MOMP) after 16 hours treatment with BRAF or MEK inhibitors (Figure 1C)": The figure legend (1C) indicates that DBP measurements were taken after 36 h drug treatment, not 16 as listed here.

5. p.7, line 24: "The effects of ZFP36 knockdown were not related to unexpected effects of siZFP36 on the MAPK pathway (Supplemental Figure 3D)." To which unexpected effects are the authors referring? There do not appear to be any obvious differences in Western blot analyses from siControl vs. siZFP36-transfected cells.

6. p.10, line 30: "we find that MAPK inhibitors leads to the decay of the pro-apoptotic factor NOXA" is inaccurate – the data in this manuscript show that MAPK inhibitors lead to decreased expression of NOXA, including ZFP36-directed destabilization of its mRNA.

7. Fig. 1F: what is designated by the dotted line in each histogram?

8. Fig. 3E and S3C: Are the ZFP36 siRNA designations consistent? (#1 and #2 in Suppl Fig. S3C but #2 and #3 in Fig. 3E).

9. Suppl Fig S4D: the right side of the Bim Western blot panel is cropped into the Bim bands

10. siRNA sequences do not appear to be listed anywhere

Reviewer #3 (Remarks to the Author):

This work by Montero et al. provides a thorough study about the role of the BH3 only protein NOXA and the antiapoptotic MCL1 in MAPK signalling dependent resistance mechanisms. According to the authors, MAPK inhibitors promote a delayed MCL1-dependent resistance, induced by the degradation of NOXA. In this scenario, the absence of the sensitizer enhances the stable binding of the available MCL1 to the BH3 only activator BIM. Moreover, they also demonstrate that MAPK inhibition destabilize NOXA via TTP/ZFP36. Finally, they suggest that the timing in the combination therapy with MAPK/MCL1 inhibitors can be critical for sensitization of cancer cells. These findings are relevant to the field, since they support a new rationale for the use of antiapoptotic inhibitors in combination with MAPK inhibitors at specific times to reverse cancer resistance.

In Fig2a, author show a WB analysing the expression levels of different proteins in A375M melanoma cells at different times upon treatment.

I) When antiapoptotics are analysed, the one that appears in the bottom part is not indicated. Is MCL1 with a different exposition or the antiapoptotic BCLXL (in coherence with Fig.4a)

II) Considering that the MCL1 dependency appears after 36h of treatment, and that after 24h it seems to be a decrease not only in NOXA but also in BAD and BCL2. I suggest to measure proteins levels at longer times 36-48h. Because one can speculate that the diminishment of BCL2 and BAD could contribute with the decrease of NOXA to enhance MCL1 dependency.

We thank the reviewers for their helpful comments and suggestions. We address all their comments below.

Reviewers' comments:

Reviewer #1 (Remarks to the Author):

In this well-conducted study, Haq and colleagues define NOXA mRNA destabilization as a common mechanism of adaptive resistance to targeted therapies. The notion that cell death induction by targeted therapies can be accentuated by treatment with BH3 mimetics is now quite established. However, even in this context, the current study offers several important insights that I believe will propel the field forward. Among these, the authors nicely demonstrate that MCL-1 is likely to be the best BCL-2 family member to target in the context of targeted therapy treatment. Technically, the work is very impressive, making use of sophisticated dynamic BH3 profiling assays conducted in a flow-based format with cell type-specific sorting, dynamic protein immunoprecipitation experiments (which are sometimes challenging to perform with BCL-2 family proteins), and beautiful *in vivo* studies with drug scheduling. Further, they articulate a mechanism by which NOXA mRNA levels are controlled by ERK-mediated phosphorylation of ZFP36. In light of these strengths, I have only minor questions and suggestions, as follows:

1. The authors nicely show that in BRAF mutant melanoma cells, NOXA decay following RAFi treatment is ZFP36-dependent, and it is likely regulated specifically by ERK-mediated ZFP36 phosphorylation at sites S218 and S228. It would be nice, if feasible, to (1) show that ZFP36 is differentially phosphorylated in the presence of MAPKi; (2) define whether ZFP36 is a direct ERK1/2 substrate; and (3) determine whether ZFP36 is required for drug-induced NOXA mRNA loss (as in Figure 3F) in other targeted therapy contexts (e.g., EGFR-, MET-, c-KIT, ALK-, and/or HER2-driven cells treated with their cognate targeted therapies).

The reviewer raises several important issues related to the regulation of ZFP36 by MAPK, however all these have been previously addressed in other manuscripts. First, he asks if ZFP36 phosphorylation is altered in the presence of MAPKi. The regulation of ZFP36 by MAPK has been described in detail by several groups. For example, Brook et al.¹ showed that ZFP36 localization and stability was regulated by MAPKs p38 and ERK through the use of various engineered mutants as well as MAPK pathway inhibitors. Bourcier et al.² showed that constitutive ERK activity downregulated ZFP36 whereas its inhibition using MEK inhibitors led to ZFP36 accumulation in melanoma cells.

The reviewer also asks about the direct interaction of MAPK with ZFP36. This also has been demonstrated by several groups. Cao et al.³ showed that ZFP36 could be phosphorylated by p42/ERK2 *in vitro*, consistent with their observation that Ser218/Ser228 were within a motif for MAPK phosphorylation⁴. Both Cao et al.⁵ and Taylor et al.⁶ later showed that recombinant ZFP36 is directly phosphorylated by three members of the MAPK family, including p42 MAPK (ERK2). Using protease digestion experiments and site-directed mutagenesis approaches, Taylor also demonstrated that Ser-220 of murine ZFP36 (homologous to human Ser-228) was phosphorylated by MAPK *in vitro*, consistent with its stimulation by mitogens in intact cells.

Third, the reviewer wonders if ZFP36 is required for drug-induced NOXA mRNA loss in other targeted therapy contexts. We agree that it is highly likely since both targeted

therapies and MAPKi induce the same change in NOXA mRNA and MCL-1 dependence in the same cells (Figure 1d).

Overall, our manuscript is consistent with the direct regulation of ZFP36 by MAPKs, as noted by many previous studies. To directly address the reviewer's comments, we have added additional text in our manuscript referencing the above studies.

2. The work on scheduling MAPKi  MCL-1i is very nice, but a few opportunities to further clarify the authors thinking should be considered. First, beyond the obvious notion that two drugs are often more toxic than one, why do the authors pursue scheduling without much attention toward simply administering the drugs in parallel? Second, in the in vivo scheduling experiments (where results are quite impressive), how exactly did the scheduling work? I found it difficult to understand exactly how the mice were dosed in these experiments. Finally, one can imagine that knowledge of the kinetics of drug-induced NOXA loss, and subsequent NOXA rebound following drug withdrawal, could directly inform the kinetics of optimal scheduling. Can the authors discuss this in more detail?

The reviewer asks us to discuss sequential dosing versus concomitant dosing of MAPKi/MCL1i. We have therefore added several sentences in our discussion addressing this interesting question.

Our motivation in exploring the sequential strategy *in vivo* was primarily based by our observation that MAPK inhibition increases MCL1 dependence *in vitro* (Figure 1c-1f, Supplemental Figure S1d-f) as well as *in vivo* (Figure 5b, Figure S5a). However, we compared sequential versus concomitant treatment strategy *in vitro* (right; Figure 4J). For this experiment, we treated cells as follows: (1) BRAFi (24h) followed by MCL1i (24h), or (2) BRAFi and MCL1i (both 48h concomitantly) (see Figure 4F for details). We observed greater synergy with sequential treatment, although there was some synergy in the concomitant strategy (this may reflect induced MCL1 dependence generated during the 48h treatment with BRAFi). One advantage of performing these experiments *in vitro* is to strictly control dose and timing (i.e., 24h) of the drug. However, interpreting such data from comparable *in vivo* studies would be challenging without specific pharmacokinetic and pharmacodynamic information regarding BRAFi and MCL1i in our mouse model. We believe that these PK/PD studies are beyond the scope of a pre-clinical study, we have added a discussion about optimal timing to our discussion section along with the caveat that these studies will need to be conducted *in vivo*. Nonetheless, as the reviewer mentioned, we believe that sequential treatment could reduce toxicity. Its application to a clinical setting will require measurement of MCL-1 dependence *in vivo*, as we have demonstrated in our pre-clinical model (Figure 5b, Figure S5a).

The reviewer asks for a complete description of the sequential treatments. These experiments are detailed under "Animal experiments" in the Methods section, but we have made described them in more detail in the figure (Figure 5c-e) and the text. We have added a diagram depicting the dosing strategy (Figure 5c).

Finally, the reviewer asks if the loss of NOXA could be a basis for optimal dosing. We agree that the kinetics of NOXA loss might provide some information about optimal sequencing strategies. However, NOXA suppression (and its rebound after MAPKi withdrawal) would be expected to be better biomarkers of ERK inhibition, rather than MCL-1 dependence. For example, we see NOXA levels decrease rapidly at both mRNA (Figure 3a) and protein (Figure 2a), yet the MCL-1 dependence follows slightly thereafter (Figure S1d). Second, the loss of NOXA is only one part of the equation, since the induced MCL-1 dependence requires distribution of MCL-1 to BIM. As our study and other previous studies have noted, BIM levels increase rapidly after MAPKi but may decrease thereafter. In sum, there are multiple variables to account for optimal scheduling. We feel that a superior tool to evaluate optimal dosing is BH3 profiling, which we performed in our pre-clinical model (Figure 5b) and can be extended to clinical contexts.

We have added a summary of these points in the Discussion section as requested by the reviewer.

3. Small point: In Discussion, Line 43, "rationale" should be spelled "rational"

Thank you for noticing this error, which has now been fixed.

Reviewer #2 (Remarks to the Author):

Experiments described in this manuscript present an innovative and compelling demonstration that oncogene-targeted therapeutic approaches induce apoptotic resistance through the anti-apoptotic protein MCL-1. This is a question of profound significance, given the inability of most targeted cancer therapies to produce sustained responses. The authors show that therapy-induced resistance is accompanied by suppression of the pro-apoptotic MCL-1-inhibitory factor NOXA in an Erk-dependent manner, at least in part via destabilization of NOXA mRNA via the RNA-binding protein ZFP36. NOXA competes for binding to MCL-1 with the anti-apoptotic factor Bim, suggesting that inhibitors of the MCL-1:Bim interaction might prove useful adjuvants to current oncogene-targeted therapies, a model supported by data from both cultured cell and tumor xenograft model systems.

Much of the study is admirably rigorous, using orthogonal approaches in multiple established cell lines, primary tumor cells, and murine xenografts for several key experiments. The impact of the work is also broadly applicable, as this acquired apoptotic resistance mechanism was observed across a range of drugs targeting different driver oncoproteins. Critical RNAi experiments included multiple independently-targeted siRNAs to ensure response specificity. Loss- and gain-of-function strategies are frequently employed, with complementary data from both in some circumstances. With a subset of experiments, however, there are concerns in presentation and/or interpretation:

1. In Figs. 3B, 3F, and Suppl. Fig. S3B, mRNA decay data should be plotted as a function of actinomycin D treatment, since this is the reagent that silences transcription (also, for one data set in each panel, actinomycin D is the only reagent added). The description of these experiments in the Methods section is confusing and does not reflect the data

presented. This is especially true for the time points listed as well as the actinomycin dose given (10 mg/ml), which is not soluble at that concentration in aqueous solution.

We agree with the reviewer and have adjusted our graphs in Figure 3b, 3f and Supplementary Figure 3b as a function of actinomycin D treatment. The reanalysis does not change the interpretation significantly or the statistical significance, since Actinomycin D was added just 30 min before the BRAF inhibitor.

We have revised the description of the Actinomycin D experiments to improve clarity. The reviewer is correct that the Actinomycin D was used at 10 μ g/mL rather than 10 mg/mL. We apologize for this typographical error.

2. Based on the data provided in Fig. 3, the contribution of accelerated mRNA decay to suppression of NOXA mRNA (and protein) levels appears to be real but minor. In Fig. 3A (and Fig. 2E), treatment with BRAF or MEK inhibitors decreased NOXA mRNA levels by 75-80% (it is also noted that the magnitude of the decrease in NOXA protein was not calculated), yet Fig. 3B only shows a 30% or so acceleration of mRNA decay. It is highly probable that other gene regulatory mechanisms, particularly transcription, play major roles in repressing NOXA expression in this system. If ZFP36-directed mRNA decay is the major mechanism accounting for suppressing NOXA following treatment with targeted therapies, then NOXA levels should be unaffected by such treatments in ZFP36-silenced cell models; however, this experiment was not reported. In the absence of this experiment, these points should be addressed in the Discussion as a likely limitation of the proposed mechanism for NOXA suppression.

We thank the reviewer for this insightful comment. We observe a 47% difference in the half-life of NOXA following BRAFi in Actinomycin D treated cells (Figure 3b). This indicates that additional mechanisms might also be involved in the regulation of the BRAFi-induced NOXA suppression. As requested by the reviewer, the silencing of ZFP36 does significantly increase the stability of the NOXA mRNA (Figure 3e).

3. The standard for validation of siRNA efficacy is to demonstrate protein suppression, as the authors do in Suppl. Fig. S2L, since validating by target mRNA level alone often overestimates knockdown efficiency for stable proteins while also ignoring potential effects of translational suppression. This concern applies to Figures S1A, S3C, and 3E.

We agree with the reviewer that validating targets by mRNA might overestimate knockdown efficiency and ignore the potential effects of translational suppression. We performed initial validation of the efficacy of the siRNAs targeting BCL2 family members by qPCR, but validated hits using Western blotting (Figure S1b). However, at the reviewers' request, we have confirmed the suppression of BCL2 family proteins by Western (Figure S1b).

Regarding validation of ZFP36 siRNAs, we have not found antibodies that are specific for endogenous TTP/ZFP36, despite trying three different vendors.

4. Statistical significance is not indicated in many figure panels, even though often indicated in the text. For example, no indication of significance is included for results of BIM treatment in Fig. 1C (or anywhere in 1E, 1F). Other examples include Figs. S1F, S2M, 3D, and 3F.

We have revised all the figures to include statistical significance as well as updated legends reflecting the specific statistical method used to test significance. To perform BH3 profiling on small quantities of freshly harvested patient samples without prolonged cell culture, these experiments were done once. We have therefore clarified the figure legend for Figure 1f.

5. In Supplemental Fig. S5B, close cropping of the Western blots for MCL-1 is partially obscuring a higher MW band appearing in both the IP and WCL lanes, particularly for drug treated tumor #1 (and possibly vehicle-treated tumor #3?). The authors should show and explain (or speculate) on the appearance of this band, particularly since it does not seem to appear in any of their other blots using the MCL-1 antibody.

We apologize that cropping of the blots has introduced some confusion in the interpretation of the blots. We have included the full, original blots in our Source Data file for the reviewer/readers and included the relevant portions below. As is seen in the blot of the whole cell lysates, we observe only one band at the appropriate size for MCL1 (red circle, lanes 19-28), showing that levels of MCL-1 do not change following dabrafenib treatment. The higher molecular weight band (~50kDa) is too large for MCL-1, therefore we believe them to be non-specific. In our immunoprecipitations, we did notice some background in the IP:Bim blot, however the higher MW band (green circle, compare lane 4 and 13) appears in the IgG control also, indicating that this band is likely non-specific and/or related to the immunoprecipitation. However, here again, we see a specific band (red circle, lane 14) that is not seen in the immunoglobulin control.

Minor points:

1. The language and grammar of select sections of the manuscript are difficult to follow, particularly in the Discussion where a few sentences make no sense as written.

We have edited the language and grammar of the Discussion and reviewed the entire manuscript, making numerous changes as noted in the revised manuscript.

2. A wide variety of inhibitors are used in this study. A supplemental table listing drugs used along with their targets would be helpful for non-experts in field.

We have added a Supplemental Table with a listing of the drugs and their targets (Supplemental Table 1b).

3. p.4 line 25: inhibitor for ERBB2 lines left out of list

We have added this to the text.

4. p.4 lines 44-47: “suppression of all BCL-2 family members (using the BIM peptide) induced a significant increase in overall mitochondrial outer membrane permeabilization (MOMP) after 16 hours treatment with BRAF or MEK inhibitors (Figure 1C)”: The figure legend (1C) indicates that DBP measurements were taken after 36 h drug treatment, not 16 as listed here.

We have revised the text to correctly state the duration of treatment of 36h.

5. p.7, line 24: “The effects of ZFP36 knockdown were not related to unexpected effects of siZFP36 on the MAPK pathway (Supplemental Figure 3D).” To which unexpected effects are the authors referring? There do not appear to be any obvious differences in Western blot analyses from siControl vs. siZFP36-transfected cells.

In this experiment, we evaluate if ZFP36 knockdown affects MAPK activation/phosphorylation. If seen, the observed effects of ZFP36 knockdown on NOXA stability could be trivially related to effects on the MAPK-regulated expression of NOXA. The reviewer is correct that we did not observe any impact of ZFP36 knockdown on basal MAPK activation, or MAPK suppression following PLX4720 treatment.

6. p.10, line 30: “we find that MAPK inhibitors leads to the decay of the pro-apoptotic factor NOXA” is inaccurate – the data in this manuscript show that MAPK inhibitors lead to decreased expression of NOXA, including ZFP36-directed destabilization of its mRNA.

We have revised this statement per the reviewers' suggestion.

7. Fig. 1F: what is designated by the dotted line in each histogram?

We have removed this dotted line from each histogram.

8. Fig. 3E and S3C: Are the ZFP36 siRNA designations consistent? (#1 and #2 in Suppl Fig. S3C but #2 and #3 in Fig. 3E).

We have verified that the ZFP36 siRNA designations are correct. We typically use two independent siRNAs for each experiment.

9. Suppl Fig S4D: the right side of the Bim Western blot panel is cropped into the Bim bands

We have corrected the cropping of this blot.

10. siRNA sequences do not appear to be listed anywhere

We have now included the specific catalog numbers of the siRNAs used in the experiments in our Methods section, for which detailed sequence and other characteristics are available online through the manufacturer.

Reviewer #3 (Remarks to the Author):

This work by Montero et al. provides a thorough study about the role of the BH3 only protein NOXA and the antiapoptotic MCL1 in MAPK signalling dependent resistance mechanisms. According to the authors, MAPK inhibitors promote a delayed MCL1-dependent resistance, induced by the degradation of NOXA. In this scenario, the absence of the sensitizer enhances the stable binding of the available MCL1 to the BH3 only activator BIM. Moreover, they also demonstrate that MAPK inhibition destabilize NOXA via TTP/ZFP36. Finally, they suggest that the timing in the combination therapy with MAPK/MCL1 inhibitors can be critical for sensitization of cancer cells. These findings are relevant to the field, since they support a new rationale for the use of antiapoptotic inhibitors in combination with MAPK inhibitors at specific times to reverse cancer resistance.

In Fig2a, author show a WB analysing the expression levels of different proteins in A375M melanoma cells at different times upon treatment.

I) When antiapoptotics are analysed, the one that appears in the bottom part is not indicated. Is MCL1 with a different exposition or the antiapoptotic BCLXL (in coherence with Fig.4a)

We apologize regarding this layout error, which has now been fixed.

II) Considering that the MCL1 dependency appears after 36h of treatment, and that after 24h it seems to be a decrease not only in NOXA but also in BAD and BCL2. I suggest to measure proteins levels at longer times 36-48h. Because one can speculate that the diminishment of BCL2 and BAD could contribute with the decrease of NOXA to enhance MCL1 dependency.

The reviewer raises the interesting possibility that diminishment of BCL-2 (via BAD) could contribute to the decrease with the loss of NOXA to enhance MCL1 dependency. Therefore, we have measured BIM dependence (BIM peptide), MCL1 dependence (NOXA peptide), BCL-2/BCL-XL dependence (BAD peptide) after 16-72hrs:

We find minimal changes in BCL-2/BCL-XL dependence over this period, whereas MCL-1 dependence increases as expected based on our manuscript. Profiling with the BIM peptide reflects an overall increase in apoptotic sensitivity (“priming”). Thus, these data are not consistent with a role of BCL-2 in the adaptive apoptotic resistance observed.

References

- 1 Brook, M. *et al.* Posttranslational regulation of tristetraprolin subcellular localization and protein stability by p38 mitogen-activated protein kinase and extracellular signal-regulated kinase pathways. *Mol Cell Biol* **26**, 2408-2418, doi:10.1128/MCB.26.6.2408-2418.2006 (2006).
- 2 Bourcier, C. *et al.* Constitutive ERK activity induces downregulation of tristetraprolin, a major protein controlling interleukin8/CXCL8 mRNA stability in melanoma cells. *Am J Physiol Cell Physiol* **301**, C609-618, doi:10.1152/ajpcell.00506.2010 (2011).
- 3 Cao, H. Expression, purification, and biochemical characterization of the antiinflammatory tristetraprolin: a zinc-dependent mRNA binding protein affected by posttranslational modifications. *Biochemistry* **43**, 13724-13738, doi:10.1021/bi049014y (2004).
- 4 Cao, H. *et al.* Identification of the anti-inflammatory protein tristetraprolin as a hyperphosphorylated protein by mass spectrometry and site-directed mutagenesis. *Biochem J* **394**, 285-297, doi:10.1042/BJ20051316 (2006).
- 5 Cao, H., Dzineku, F. & Blackshear, P. J. Expression and purification of recombinant tristetraprolin that can bind to tumor necrosis factor-alpha mRNA and serve as a substrate for mitogen-activated protein kinases. *Arch Biochem Biophys* **412**, 106-120 (2003).
- 6 Taylor, G. A., Thompson, M. J., Lai, W. S. & Blackshear, P. J. Phosphorylation of tristetraprolin, a potential zinc finger transcription factor, by mitogen stimulation in intact cells and by mitogen-activated protein kinase in vitro. *The Journal of biological chemistry* **270**, 13341-13347 (1995).

REVIEWERS' COMMENTS:

Reviewer #1 (Remarks to the Author):

No further questions. Great work!

Reviewer #2 (Remarks to the Author):

This manuscript describes an impressive body of work that defines a common molecular mechanism that limits the effectiveness of several oncogene-targeted chemotherapeutics. Previous criticisms from the reviewers were relatively minor. The authors complied with most concerns made by reviewers and suitably addressed those that could not be accommodated for practical reasons, with a few small exceptions:

1. Horizontal axis for graph in Supplemental Fig. 3b still requires re-labeling (should be actinomycin D)
2. siRNA sources for MCL1 and BCL2 family members have been included, but still missing siRNA sources for ERK1/2 (used in Supplemental Figs. 2l and 2m), a large number of BH3 proteins (used in Supplemental Fig. 2n), and the ZFP36 family members (used in Fig. 3 and Supplemental Figure 3).

Reviewer #3 (Remarks to the Author):

The authors have addressed the reviewer's concerns adequately.

Response to Review Comments

We thank all reviewers for their enthusiasm about our manuscript and for their comments that have generated an improved manuscript.

Reviewer #2 (Remarks to the Author):

This manuscript describes an impressive body of work that defines a common molecular mechanism that limits the effectiveness of several oncogene-targeted chemotherapeutics. Previous criticisms from the reviewers were relatively minor. The authors complied with most concerns made by reviewers and suitably addressed those that could not be accommodated for practical reasons, with a few small exceptions:

1. Horizontal axis for graph in Supplemental Fig. 3b still requires re-labeling (should be actinomycin D)

We have edited Supplemental Figure 3b as requested.

2. siRNA sources for MCL1 and BCL2 family members have been included, but still missing siRNA sources for ERK1/2 (used in Supplemental Figs. 2l and 2m), a large number of BH3 proteins (used in Supplemental Fig. 2n), and the ZFP36 family members (used in Fig. 3 and Supplemental Figure 3).

We have updated the manuscript to include sources for all the siRNAs used.